# Allogeneic CD33-directed CAR-NKT cells for the treatment of bone marrow-resident myeloid malignancies

Yan-Ruide Li [1,2,12], Ying Fang[1,2,12], Siyue Niu[1,2,12], Yichen Zhu[1,2,12], Yuning Chen [1,2], Zibai Lyu[1,2], Enbo Zhu [1,3], Yanxin Tian[1,2], Jie Huang[1,2], Valerie Rezek [4,5,6], Scott Kitchen [4,5,6], Tzung Hsiai [3], Jin J. Zhou[7], Pin Wang [8], Wanxing Chai-Ho[4], Sunmin Park[4], Christopher S. Seet [4,5,9,10], Caspian Oliai[4] & Lili Yang [1,2,5,9,10,11] ✉

Chimeric antigen receptor (CAR)-engineered T cell therapy holds promise for treating myeloid malignancies, but challenges remain in bone marrow (BM) infiltration and targeting BM-resident malignant cells. Current autologous CAR-T therapies also face manufacturing and patient selection issues, underscoring the need for off-the-shelf products. In this study, we characterize primary patient samples and identify a unique therapeutic opportunity for CAR-engineered invariant natural killer T (CAR-NKT) cells. Using stem cell gene engineering and a clinically guided culture method, we generate allogeneic CD33-directed CAR-NKT cells with high yield, purity, and robustness. In pre-clinical mouse models, CAR-NKT cells exhibit strong BM homing and effectively target BM-resident malignant blast cells, including CD33-low/negative leukemia stem and progenitor cells. Furthermore, CAR-NKT cells synergize with hypomethylating agents, enhancing tumor-killing efficacy. These cells also show minimal off-tumor toxicity, reduced graft-versus-host disease and cytokine release syndrome risks, and resistance to allorejection, highlighting their substantial therapeutic potential for treating myeloid malignancies.

Myeloid malignancies, such as acute myeloid leukemia (AML) and myelodysplastic syndrome (MDS), are primarily diseases of the bone marrow (BM), where they interfere with normal hematopoiesis[1–4]. The standard treatment for these diseases involves aggressive chemotherapy utilizing agents such as daunorubicin and cytarabine, followed by allogeneic stem cell transplantation[5,6]. Additionally, new therapeutic modalities, including hypomethylating agents (HMAs), monoclonal antibodies, and cancer growth inhibitors, have been made accessible[7–10]. Despite these advancements, the 5-year survival rates for AML and MDS remain dismal at 37% and 30%, respectively[1–4]. A major contributor to therapeutic resistance is the persistence of malignant blast cells in the BM, particularly leukemia stem and progenitor cells

[1]Department of Microbiology, Immunology & Molecular Genetics, University of California, Los Angeles, CA, USA. [2]Department of Bioengineering, University of California, Los Angeles, CA, USA. [3]Division of Cardiology, Department of Medicine, David Geffen School of Medicine, University of California, Los Angeles, CA, USA. [4]Division of Hematology-Oncology, Department of Medicine, David Geffen School of Medicine, University of California, Los Angeles, CA, USA. [5]The Eli and Edythe Broad Center of Regenerative Medicine and Stem Cell Research, University of California, Los Angeles, CA, USA. [6]UCLA AIDS Institute, David Geffen School of Medicine, University of California, Los Angeles, CA, USA. [7]Department of Biostatistics, Fielding School of Public Health, University of California, Los Angeles, CA, USA. [8]Department of Chemical Engineering and Materials Science, University of Southern California, Los Angeles, CA, USA. [9]Jonsson Comprehensive Cancer Centre, University of California, Los Angeles, CA, USA. [10]Molecular Biology Institute, University of California, Los Angeles, CA, USA. [11]Parker Institute for Cancer Immunotherapy, University of California, Los Angeles, CA, USA. [12]These authors contributed equally: Yan-Ruide Li, Ying Fang, Siyue Niu, Yichen Zhu. ✉e-mail: liliyang@ucla.edu

(LSPCs), a subpopulation characterized by their self-renewal capacity and disease propagation propensity[11,12]. The BM microenvironment plays a pivotal role in supporting LSPC survival and proliferation, largely through signaling pathways and metabolic reprogramming[13–15]. Moreover, specific cues from the BM niche promote LSPC quiescence, which shields these cells from chemotherapy-induced cytotoxicity[16,17]. This underscores the critical need for developing innovative therapeutic approaches aimed at effectively targeting BM-resident LSPCs, addressing the significant unmet needs of patients with myeloid malignancies.

Chimeric antigen receptor (CAR)-engineered T (CAR-T) cell therapy has emerged as a promising treatment modality for various hematologic malignancies and solid tumors, including myeloid malignancies[18,19]. Among the identified potential therapeutic targets such as CD33, CD123, CD70, Fms-like tyrosine kinase 3 (FLT-3), and C-type lectin-like molecule-1 (CLL1), CD33 stands out as a promising target for treating myeloid malignancies[20–27]. Firstly, CD33 is prominently expressed on malignant progenitor and myeloid cells, with notable prevalence observed in AML (-80%) and MDS (-75%)[28,29]. Secondly, clinical trials have demonstrated the feasibility, safety, and efficacy of autologous CD33-directed CAR-T therapy, showing promising results in targeting relapsed/refractory AML in both pediatric and adult populations, as well as high-risk MDS in elderly individuals[30–33]. However, conventional CAR-T therapy often achieves only transient reductions in blast cells and limited anti-leukemia efficacy, which may be due to the failure of CAR-T cells to efficiently infiltrate the bone marrow and target LSPCs[20,34,35]. Moreover, CAR-T cell therapy typically relies on autologous approaches, where T cells are sourced from cancer patients, undergo intricate manufacturing processes, and are subsequently reintroduced, posing significant challenges[36–40]. This method is time-consuming, labor-intensive, and costly. Additionally, its feasibility may be limited in patients with insufficient healthy T cells or rapidly advancing cancer, potentially leading to missed therapeutic opportunities[41–43]. Consequently, there is a pressing need for an off-the-shelf CAR-engineered cell product that circumvents these limitations.

Recently, we developed a clinically guided culture method to generate allogeneic invariant natural killer T (iNKT or NKT) cells and their CAR-engineered derivatives (denoted as CAR-NKT cells) by integrating NKT TCR engineering into human hematopoietic stem and progenitor cells (HSPCs), followed by their differentiation into mature NKT cells through an ex vivo feeder-free culture[44]. NKT cells are particularly promising for targeting myeloid malignancies, as their semi-invariant TCR recognizes the non-polymorphic MHC class I-like molecule CD1d, which is abundantly expressed on malignant myeloid cells in patients with myeloid malignancies[45–47]. Notably, HMAs, which are FDA-approved therapies for myeloid malignancies, have been shown to upregulate CD1d expression on tumor cells[48], suggesting potential synergistic effects when combined with NKT cell therapies. Moreover, compared to conventional CAR-T cells, allogeneic CAR-NKT cells exhibit distinct biodistribution patterns in vivo[44], which is likely attributable to their unique chemokine receptor expression profile.

In this study, we successfully generate allogeneic CD33-directed CAR-NKT (CAR33-NKT) cells with high yield and purity using the clinically guided culture method. We conduct a comprehensive assessment of the therapeutic potential of allogeneic CAR33-NKT cells using AML and MDS using primary patient samples, humanized mouse models, and patient-derived xenograft (PDX) models. Through a series of in vitro and in vivo immunological assays combined with transcriptomics analyses, we evaluate allogeneic CAR33-NKT cell manufacturing, phenotype and functionality, mechanism of action, pharmacokinetics and pharmacodynamics (PK/PD), efficacy, safety, and immunogenicity. Notably, we assess the BM homing ability of allogeneic CAR33-NKT cells and their capacity to target BM-resident malignant blast cells, particularly LSPCs, to demonstrate a safe and effective approach for the treatment of myeloid malignancies.

## Results

### Biomarker and transcriptome profiling of primary AML and MDS samples reveal potential for CAR-NKT cell therapy

To profile myeloid malignancies and identify optimal therapeutic approaches, we performed an in-depth analysis of primary AML and MDS BM samples using flow cytometry and single-cell RNA sequencing (scRNA-seq) (Fig. 1a). We collected 8 primary AML and MDS samples, which were subjected to flow cytometry for surface biomarker characterization (Supplementary Table 1). Additionally, we analyzed two public scRNA-seq datasets involving primary AML and MDS samples to examine their transcriptomic profiles[49,50].

Four distinct subpopulations emerged throughout the disease course: leukemia stem cells (LSCs), multipotent progenitor cells (MPPs), common myeloid progenitors (CMPs), and myeloblast cells (MBCs) (Fig. 1b). Notably, the combined presence of LSCs and MPPs was denoted as LSPCs. As the disease advanced from LSPCs to CMPs to MBCs, there was a consistent upregulation of CD38 expression accompanied by a downregulation of CD34 and stem markers, which aligns with established patterns of LSPC differentiation trajectories observed in patients with myeloid malignancies[51,52].

Flow cytometry analysis of primary AML and MDS blast cells revealed three distinct subpopulations (i.e., LSPCs, CMPs, and MBCs), distinguished by the expression of CD34 and CD38 surface markers (Supplementary Fig. 1a). These subpopulations exhibited differential expression of surface tumor antigens, including the CAR target (i.e., CD33), the NKT TCR target (i.e., CD1d), and NKR ligands (i.e., CD112, CD155, and MICA/B) (Fig. 1c–e). As the disease advanced from LSPCs to CMPs to MBCs, there was a notable upregulation of CD33 and CD1d, whereas NKR ligands showed a downregulation (Fig. 1c–e). LSPCs, characterized by a more stem-like, therapy-resistant phenotype, exhibited low expression levels of CD33 and CD1d, which are typically found on myeloid progenitors and mature myeloid cells[11,12,28,29,53]. Consequently, LSPCs may evade targeting by CAR- and NKT TCR-based cell therapies. However, their high expression of NKR ligands (i.e., CD112, CD155, and MICA/B) may render them susceptible to NKR-mediated killing (Fig. 1c–e). Of note, the expression of these NK ligands was not consistent across LSPCs from different AML and MDS patient samples, suggesting variable susceptibility to NKR-mediated killing among patients (Fig. 1e). In contrast, the more differentiated CMPs and particularly MBCs demonstrated elevated levels of CD33 and CD1d (Fig. 1c, e), making them susceptible to killing by CAR- and NKT TCR-mediated mechanisms. Consequently, CAR-NKT cells present a promising therapeutic strategy for targeting AML and MDS blast cells, capitalizing on the triple tumor-targeting mechanisms of CAR, NKT TCR, and NKR.

Single-cell RNA sequencing analysis of primary patient blast cells revealed four subpopulations (i.e., LSCs, MPPs, CMPs, and MBCs), distinguished by the expression of *CD34*, *CD38*, and stem cell-related genes *SOX4* and *CD99* (Figs. 1f, g and Supplementary Fig. 1b)[49,54]. Different patient blast cells contained varying proportions of these subpopulations, reflecting diverse conditions and stages of the disease (Fig. 1h). Notably, LSCs expressed elevated levels of a cancer stem cell (CSC) gene signature, supporting their stemness phenotype and self-renewal capabilities (Fig. 1i, Supplementary Fig. 1c, and Supplementary Table 2). Intriguingly, LSCs also demonstrated heightened expression of NKR ligand gene signatures (Figs. 1j, k, Supplementary Fig. 1d, and Supplementary Table 2), consistent with flow cytometry findings (Fig. 1c, e). These observations suggest that while LSCs possess inherent resilience[11,12], they remain susceptible to NKR-mediated killing. In summary, the comprehensive biomarker and transcriptome profiling of primary AML and MDS samples uncover potential avenues for CAR-NKT cell therapy.

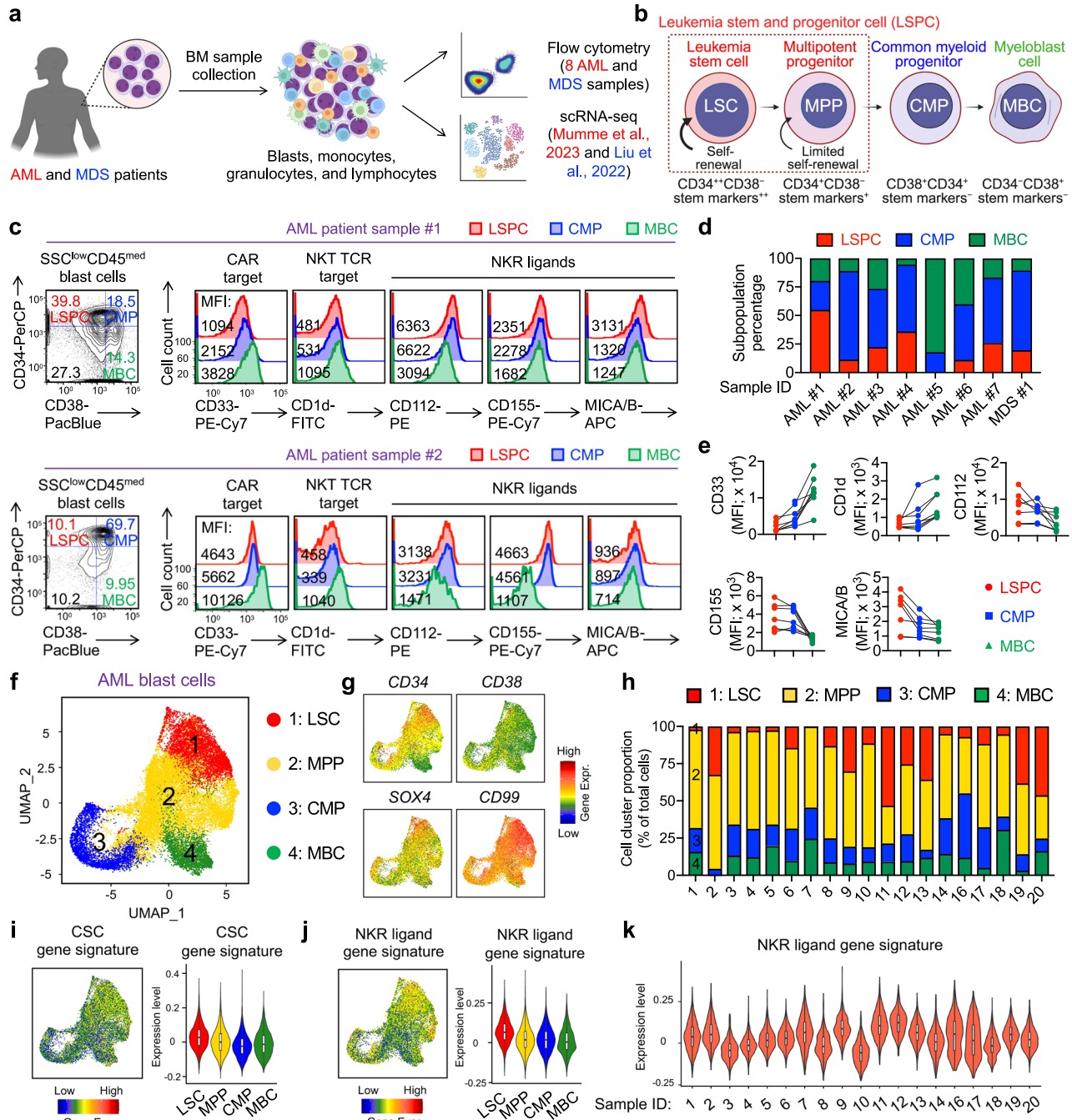

**Fig. 1 | Biomarker and transcriptome profiling of primary AML and MDS patient samples.** **a** Experiment design to profile primary AML and MDS patient bone marrow (BM) samples using flow cytometry and single cell RNA sequencing (scRNA-seq). 8 AML and MDS primary samples were included for flow cytometry analyses. Data from Gene Expression Omnibus database (GSE235923)[49] and NCBI Sequence Read Archive (PRJNA720840)[50] were included for scRNA-seq analyses. Created in BioRender. LI, Y. (2025) https://BioRender.com/o37h997. **b** Diagram showing the progression from leukemia stem cells (LSCs) to myeloblast cells (MBCs) in myeloid malignancies, along with associated biomarkers. Created in BioRender. FANG, Y. (2025) https://BioRender.com/f57l159 **c**–**e** Profiling AML and MDS blast cells using flow cytometry. **c** FACS detection of the subpopulations of AML blast cells, and their expression of CAR target (CD33), NKT TCR target (CD1d), and NKR ligands (i.e., CD112, MICA/B, and ULBP-1). Two representative data sets from AML samples #1 and #2 are presented. **d** Quantification of the proportions of the three subpopulations of AML and MDS blast cells. The combined percentage of these subpopulations totals 100%. **e** Quantification of the expression of CAR target,

NKT TCR target, and NKR ligands on the three subpopulations of AML and MDS blast cells (n = 7 for LSPC, and n = 8 for CMP and MBC; n represents different patient samples). **f**–**k** Profiling AML blast cells using scRNA-seq. Data from Gene Expression Omnibus database (GSE235923) were analyzed. **f** Combined UMAP plot showing the formation of four major cell clusters. 19 primary AML blast samples were analyzed. **g** UMAP plots showing the expression distribution of the *CD34*, *CD38*, and stem genes *SOX4* and *CD99*. **h** Bar graphs showing the cell cluster proportions of the 19 primary AML blast samples. Expression of cancer stem cell (CSC) gene signature (**i**) and NKR ligand gene signature (**j**) in the indicated cell clusters. UMAP plots showing the gene expression distributions and violin plots showing the gene expression levels are presented. Data from the 19 primary AML blast samples are shown. **k** Violin plots showing the expression distribution of NKR ligand gene signature in the 19 primary AML blast samples. Representative of 1 (**d**–**k**) and 8 (**c**) experiments. In the violin plots (**i**, **j**), box and whisker plots exhibit the minimum, lower quartile, median, upper quartile and maximum expression levels of each type of cell. Source data and exact *p* values are provided as a Source Data file.

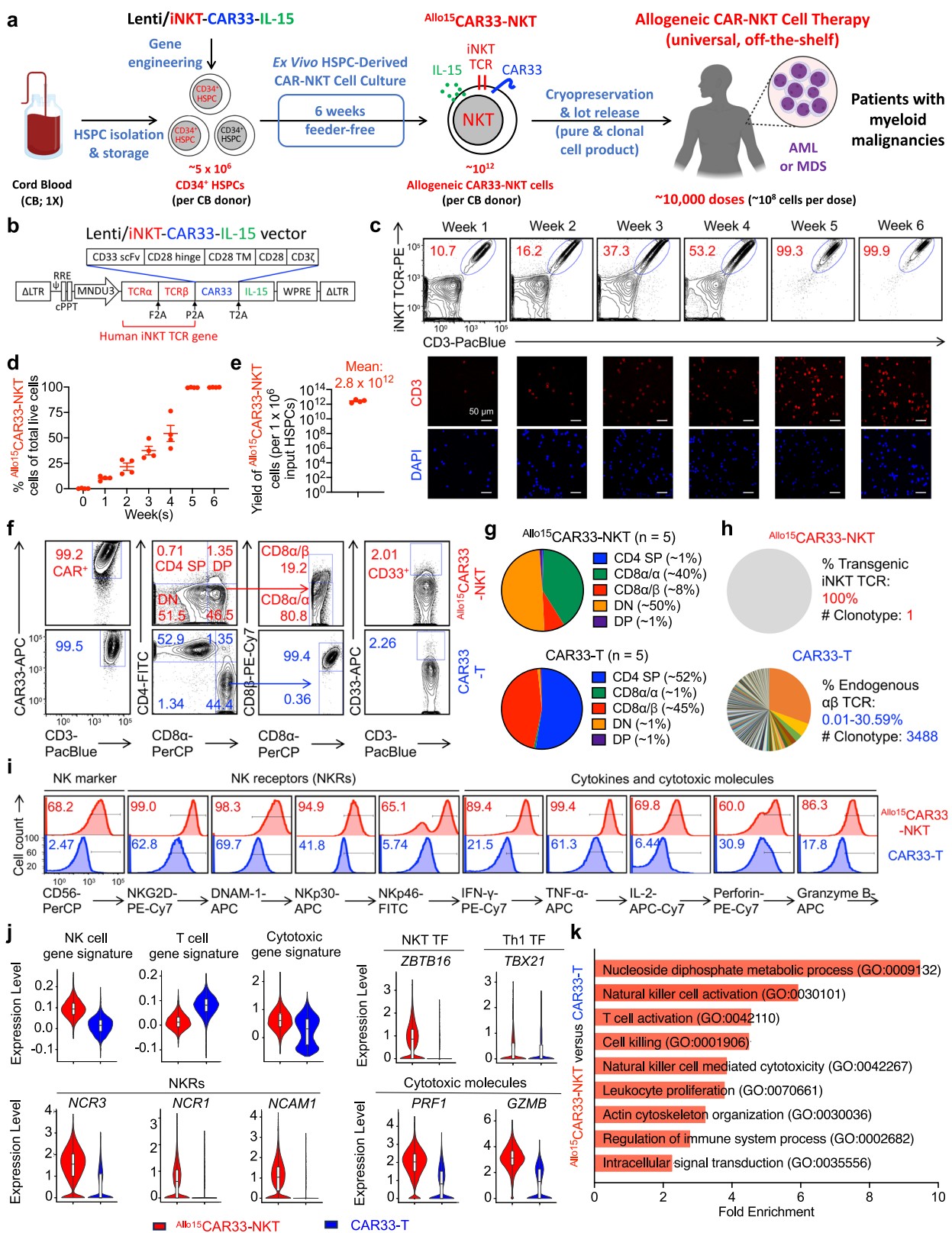

**HSPC-engineered allogeneic CAR33-NKT cells can be generated at high yield and purity using a clinically guided culture method**
We employed a previously established method to produce allogeneic CAR33-NKT cells by combining gene engineering of human HSPCs with a clinically guided culture method (Fig. 2a)[44]. We sourced CD34+ HSPCs from human cord blood (CB) from commercial suppliers such as HemaCare. These HSPCs were then cultured in a defined 5-stage,

6-week Ex Vivo HSPC-Derived CAR-NKT Cell Culture to derive allogeneic IL-15-enhanced CD33-directed CAR-NKT (Allo15CAR33-NKT) cells (Fig. 2a and Supplementary Fig. 2).

At Stage 0, freeze-thawed HSPCs were transduced with lentiviral vectors encoding genes for an iNKT TCR, CAR33, and soluble human IL-15 (Fig. 2b). The iNKT TCR gene encodes a pair of iNKT TCR α and β chains, previously used to develop autologous and allogeneic NKT cell

**Fig. 2 | Generation and characterization of HSPC-engineered allogeneic IL-15-enhanced CD33-directed CAR-NKT ($^{Allo15}$CAR33-NKT) cells. a** Schematics showing the generation of $^{Allo15}$CAR33-NKT cells. HSPC, hematopoietic stem and progenitor cells; Lenti/iNKT-CAR33-IL-15, lentiviral vector encoding a pair of iNKT TCR α and β chains, a CD33-directed CAR, and a human soluble IL-15. Created in BioRender. LI, Y. (2025) https://BioRender.com/o37h997. **b** Schematics showing the design of Lenti/iNKT-CAR33-IL-15 lentivector. ΔLTR, self-inactivating long terminal repeats; MNDU3, internal promoter derived from the MND retroviral LTR U3 region; φ, packaging sequence; RRE, rev-responsive element; cPPT, central polypurine tract; WPRE, woodchuck hepatitis virus posttranscriptional regulatory element; F2A, foot-and-mouth disease virus 2 A; P2A, porcine teschovirus-1 2A; T2A, thosea asigna virus 2A. **c** FACS and immunofluorescence (IF) monitoring of the generation of $^{Allo15}$CAR33-NKT cells during the 6-week culture. iNKT TCR was stained using a 6B11 monoclonal antibody. **d** Percentage of $^{Allo15}$CAR33-NKT cells in total live cells during the 6-week culture ($n = 4$; n indicates different CB donors). **e** Yield of $^{Allo15}$CAR33-NKT cells ($n = 4$; n indicates different CB donors). **f** FACS detection of surface

markers on $^{Allo15}$CAR33-NKT cells. Healthy donor peripheral blood mononuclear cell (PBMC)-derived conventional CD33-directed CAR-engineered T (CAR33-T) cells were included as a control. DN double-negative, DP double-positive. **g** Comparison of the indicated subpopulation percentages between $^{Allo15}$CAR33-NKT and conventional CAR33-T cells ($n = 5$; n indicates different cell batches) **h** Single cell TCR sequencing analyses of $^{Allo15}$CAR33-NKT and conventional CAR33-T cells. **i** FACS detection of NK marker and NK receptor (NKR) expression, as well as intracellular cytokine and cytotoxic molecule production of $^{Allo15}$CAR33-NKT and conventional CAR33-T cells. **j** Violin plots showing the expression distribution of the indicated gene signatures in $^{Allo15}$CAR33-NKT and conventional CAR33-T cells. TF, transcription factor. **k** Pathway analyses of differentiated expressed genes comparing $^{Allo15}$CAR33-NKT with conventional CAR33-T cells. GO, Gene ontology ID. Representative of 1 (**h**, **j**, **k**) and >5 (**a–g**, **i**) experiments. For the scTCR-seq (**h**) and scRNA-seq analyses (**j**, **k**), one $^{Allo15}$CAR33-NKT sample (containing 12,006 cells) and one CAR33-T sample (containing 9122 cells) were analyzed. Source data and exact $p$ values are provided as a Source Data file.

therapies for cancer treatment[55,56]. The CAR33 targets CD33, a surface antigen found on most blast cells in patients with myeloid malignancies[57]. IL-15 is known to improve the in vivo persistence and antitumor efficacy of CAR-NKT cells, as demonstrated in preclinical studies and clinical trials[58–60]. The lentiviral vector showed efficient transduction of all CB HSPCs tested, consistently achieving a transduction rate over 50% (Supplement Fig. 3a, b). The gene-engineered HSPCs were cultured in a classical X-VIVO 15-based serum-free HSPC medium for 48 h (Fig. 2a and Supplementary Fig. 2).

These HSPCs were subsequently cultured ex vivo in a scalable 6-week protocol to generate $^{Allo15}$CAR33-NKT cells: Stage 1 HSPC expansion (~2 weeks), Stage 2 NKT differentiation (~1 week), Stage 3 NKT deep differentiation (~1 week), and Stage 4 NKT expansion (~2 weeks) (Fig. 2a and Supplementary Fig. 2). The Stage 1 Culture Medium comprised the StemSpan™ SFEM II Medium (SFEM) and the StemSpan™ Lymphoid Progenitor Expansion Supplement to support the HSC expansion[61,62]. The Stage 2 Culture Medium comprised the SFEM and the StemSpan™ Lymphoid Progenitor Maturation Supplement (LPMS) to support the NKT cell differentiation[63]. The Stage 3 Culture Medium comprised the SFEM, the LPMS, the CD3/CD28/CD2 T Cell Activator, and the human recombinant IL-15 to support NKT cell deep differentiation[63]. In addition, the StemSpan™ Lymphoid Differentiation Coating Material were utilized throughout Stages 1 to 3 to support HSPC expansion and differentiation into T cell lineage.

At Stage 4, three NKT cell expansion methods were employed: a feeder-free, serum-free approach using αCD3/αCD28 antibodies, and two feeder-dependent methods utilizing α-galactosylceramide (αGC)-loaded healthy donor peripheral blood mononuclear cells (PBMCs) or CD33-overexpressing artificial antigen-presenting cells (aAPCs) (Fig. 2a and Supplementary Fig. 2). All three expansion strategies have been demonstrated to be suitable for clinical and commercial development of cell products[42,43].

Both flow cytometry and immunofluorescence imaging confirmed the successful generation of $^{Allo15}$CAR33-NKT cells (Fig. 2c and Supplementary Fig. 3c–f). The proportion of NKT cells increased progressively each week, starting from approximately 10% in the first week and reaching over 99% by the sixth week (Fig. 2c, d and Supplementary Fig. 3f). This process resulted in a high yield and high purity of $^{Allo15}$CAR33-NKT cells, with almost 100% expression of CAR33, attributable to the presence of the iNKT TCR and CAR33 in the same lentiviral vector (Fig. 2e and Supplementary Fig. 3c–e). From a single CB donor containing approximately ~$5 \times 10^6$ CD34$^+$ HSPCs, an estimated ~$10^{12}$ mature $^{Allo15}$CAR33-NKT cells could be produced (Fig. 2e). Given the typical dose of $10^8$–$10^9$ cells per treatment in current autologous CAR-T cell therapy[64], the $^{Allo15}$CAR33-NKT cells could potentially be distributed into ~1000–10,000 doses for treating cancer patients (Supplementary Fig. 3e).

Besides the one-lentivector system, $^{Allo15}$CAR33-NKT cells could also be generated using a two-lentivector system (Supplementary Fig. 4a). In this approach, both Lenti/iNKT-IL-15 and Lenti/CAR33 lentivectors were used simultaneously to transduce HSPCs (Supplementary Fig. 4b). This resulted in the formation of four distinct HSPC populations: iNKT TCR$^+$CAR33$^+$ HSPCs, which differentiated into $^{Allo15}$CAR33-NKT cells; iNKT TCR$^+$CAR33$^-$ HSPCs, which differentiated into allogeneic IL-15-enhanced NKT ($^{Allo15}$NKT) cells; and iNKT TCR$^-$CAR33$^+$ and iNKT TCR$^-$CAR33$^-$ HSPCs, which did not differentiate into mature NKT cells (Supplementary Fig. 4c). During Stage 1–3 culture, both $^{Allo15}$CAR33-NKT and $^{Allo15}$NKT cells were successfully generated, maintaining about 1:1 ratio (Supplementary Fig. 4d). At stage 4, the resulting cell population was stimulated with CD33-overexpressing aAPCs, which selectively enriched $^{Allo15}$CAR33-NKT cells and ultimately yielded a purity of >95% in the $^{Allo15}$CAR33-NKT cell population (Supplementary Fig. 4e, f). Both one-lentivector and two-lentivector systems generated $^{Allo15}$CAR33-NKT cells with comparable cell yield and purity, demonstrating the robustness of the ex vivo feeder-free culture system for the production of these engineered cells (Supplementary Fig. 4g–j).

Next, the phenotype and functionality of $^{Allo15}$CAR33-NKT cells were evaluated alongside healthy donor PBMC-derived conventional CAR33-engineered T (CAR33-T) cells, which served as a benchmark control (Fig. 2f and Supplementary Fig. 5a–e). Although CAR33 expression was typically around 80% in CAR33-T cells (Supplementary Fig. 5b), manufacturing variations and purification processes might impact the consistency of CAR33-T cell production. In contrast, $^{Allo15}$CAR33-NKT cells achieved near 100% CAR33 expression (Fig. 2f), which ensures that each cell is armed with the CAR and negates the need for additional purification or CAR enrichment steps.

Flow cytometry analysis revealed that $^{Allo15}$CAR33-NKT cells exhibited a CD8 single-positive (SP) and double-negative (DN) phenotype (Fig. 2f, g and Supplementary Fig. 5f), notably lacking a CD4 SP population commonly observed in endogenous human NKT cells as well as CAR33-T cells[65–67]. Furthermore, the CD8$^+$ $^{Allo15}$CAR33-NKT cells predominantly expressed the CD8α/α isoform, with a lower prevalence of the CD8α/β isoform, mirroring the pattern observed in endogenous human NKT cells and differing from the profile seen in CAR33-T cells (Fig. 2f, g and Supplementary Fig. 5f). Functionally, the CD8 SP (particularly CD8α/α) and DN $^{Allo15}$CAR33-NKT cells displayed similar characteristics, suggesting a pro-inflammatory and highly cytotoxic profile, which is advantageous for cancer immunotherapy[68–71]. However, CD4 SP NKT cells also possess unique therapeutic potential, notably in immune regulation[67]. Modifications to the culture protocol could potentially enable the generation of a CD4 SP $^{Allo15}$CAR33-NKT cell population. Moreover, neither $^{Allo15}$CAR33-NKT nor CAR33-T cells expressed CD33, indicating they would not be

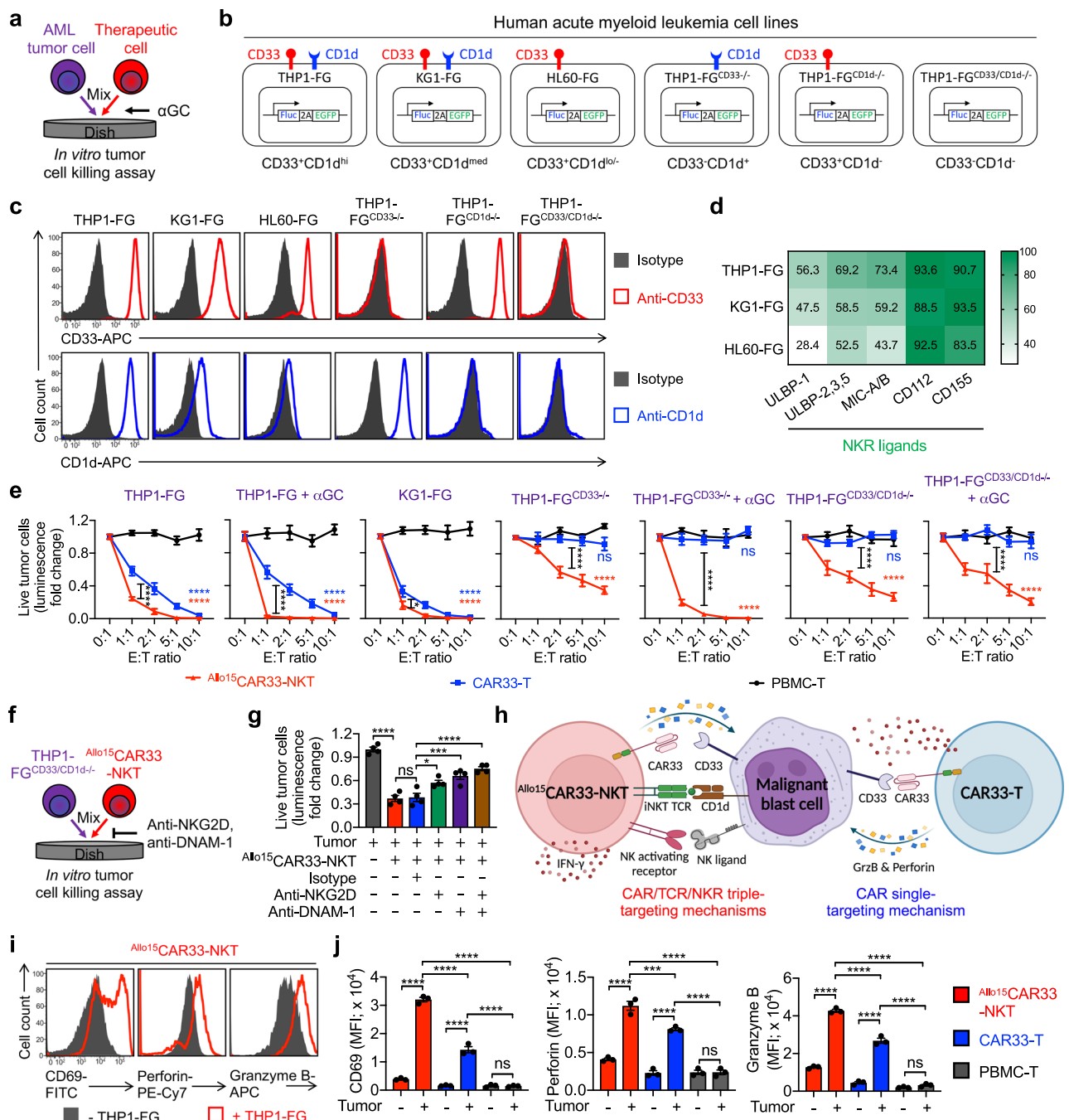

**Fig. 3 | In vitro tumor targeting efficacy and mechanisms of <sup>Allo15</sup>CAR33-NKT cells. a–e** Studying the in vitro antitumor efficacy of <sup>Allo15</sup>CAR33-NKT cells against human AML cell lines. CAR33-T cells and non-CAR33-engineered PBMC-T cells were included as therapeutic cell controls. **a** Experimental design. **b** Schematics showing the indicated human AML cell lines. THP1-FG, THP1 cell line engineered to overexpress the firefly luciferase and green fluorescence protein dual reporters (FG); KG1-FG, KG1 cell line engineered to overexpress FG; HL60-FG, HL60 cell line engineered to overexpress FG; THP1-FG<sup>CD33-/-</sup>, THP1-FG cell line further engineered to knockout the *CD33* gene; THP1-FG<sup>CD1d-/-</sup>, THP1-FG cell line further engineered to knockout the *CD1d* gene; THP1-FG<sup>CD33/CD1d-/-</sup>, THP1-FG cell line further engineered to knockout the *CD33* and *CD1d* genes. **c** FACS detection of CD33 and CD1d expressions on the indicated AML cells. **d** Heatmap showing the NKR ligand expressions on the indicated AML cells. The number represents the percentage of NKR ligand-positive tumor cells out of the total tumor cells. Three independent tumor cell samples were analyzed, and the average numbers are presented.

**e** Tumor cell killing data at 24 h (*n* = 4 from four different cell product donors). **f, g** Studying the tumor cell killing mechanisms of <sup>Allo15</sup>CAR33-NKT cells mediated by NKRs (i.e., NKG2D and DNAM-1). **f** Experimental design. **g** Tumor cell killing data at 24 h (E:T ratio = 10:1; *n* = 4 from four different cell product donors). **h** Diagram showing the CAR/TCR/NKR triple tumor-targeting mechanisms of <sup>Allo15</sup>CAR33-NKT cells, and the CAR single tumor-targeting mechanism of CAR33-T cells. GrzB, Granzyme B. Created in BioRender. FANG, Y. (2025) https://BioRender.com/j50y057. **i, j** Studying the expression of effector molecules of <sup>Allo15</sup>CAR33-NKT cells. **i** FACS detection of surface CD69 as well as intracellular Perforin and Granzyme B in <sup>Allo15</sup>CAR33-NKT cells. **j** Quantification of (**i**) (*n* = 3 from three different cell product donors). Representative of 3 experiments. Data are presented as the mean ± SEM. ns not significant, \**p* < 0.05; \*\**p* < 0.01; \*\*\**p* < 0.001; \*\*\*\**p* < 0.0001, by one-way ANOVA (**g, j**), or two-way ANOVA (**e**). Source data and exact *p* values are provided as a Source Data file.

targeted by the CAR33, thereby reducing the risk of fratricide (Fig. 2f and Supplementary Fig. 5c).

Both flow cytometry and single-cell TCR sequencing analyses confirmed the uniform expression of transgenic iNKT TCRs in [Allo15]CAR33-NKT cells (Fig. 2f, h). Randomly recombined endogenous αβ TCRs were not detected, likely due to allelic exclusion induced by the overexpression of transgenic iNKT TCRs in HSPCs[72,73]. This absence of endogenous αβ TCRs minimizes the risk of graft-versus-host disease (GvHD), allowing for streamlined manufacturing of [Allo15]CAR33-NKT cells without additional purification steps.

## Allogeneic CAR33-NKT cells demonstrate prominent NK-like properties and high cytotoxic functionality

Flow cytometry analysis of [Allo15]CAR33-NKT cells revealed a typical phenotype resembling that of endogenous NKT cells, characterized by notable NK-like properties and cytotoxic functionality[74,75]. In comparison to conventional CAR33-T cells, [Allo15]CAR33-NKT cells exhibited elevated expression levels of NK marker CD56 and NKRs such as NKG2D, DNAM-1, NKp30, and NKp46 (Fig. 2i and Supplementary Fig. 5g). Additionally, [Allo15]CAR33-NKT cells demonstrated heightened production of effector cytokines such as IFN-γ, TNF-α, and IL-2, as well as cytotoxic molecules such as Perforin and Granzyme B (Fig. 2i and Supplementary Fig. 5h). These phenotypic and functional attributes, consistent with the CD8 SP/DN phenotype of [Allo15]CAR33-NKT cells, are advantageous for their potential application in cancer therapy. Notably, [Allo15]CAR33-NKT cells generated using three different expansion approaches (i.e, αCD3/αCD28 Ab, αGC/PBMCs, and aAPCs) exhibited comparable phenotypes and functionalities, indicating that the three approaches yield functional cells suitable for subsequent studies (Supplementary Fig. 6a–c).

To gain insights into the genomic and molecular characteristics of [Allo15]CAR33-NKT cells, we performed scRNA-seq analysis, using conventional CAR33-T cells as a benchmark control. [Allo15]CAR33-NKT cells exhibited a mixed T/NK cell phenotype, with a higher NK cell signature compared to CAR33-T cells and a comparatively lower T cell signature (Fig. 2j). Additionally, [Allo15]CAR33-NKT cells showed elevated expression of cytotoxic genes, such as those encoding Perforin and Granzyme B, aligning with flow cytometry results (Fig. 2i, j). Furthermore, [Allo15]CAR33-NKT cells expressed high levels of the NKT transcription factor *ZBTB16* (encoding PLZF)[76] and the Th1 transcription factor *TBX21* (encoding T-bet) (Fig. 2j)[77]. Pathway analyses confirmed that, compared to CAR33-T cells, [Allo15]CAR33-NKT cells exhibited enhanced NK-related activation and cytotoxicity, T cell activation, and proliferation gene profiles (Fig. 2k). These findings suggest that [Allo15]CAR33-NKT cells possess potent T/NK-related cytotoxic features, supporting their potential for antitumor applications.

## Allogeneic CAR33-NKT cells directly kill AML and MDS tumor cells at high efficacy and using multiple targeting mechanisms

[Allo15]CAR33-NKT cells are expected to target myeloid malignancy tumor cells through multiple surface receptors, including CAR33-mediated recognition of CD33, NKT TCR-mediated recognition of CD1d, and NKR-mediated recognition of NK ligands. This multi-targeting approach may enhance the ability of [Allo15]CAR33-NKT cells to overcome tumor immune evasion, a challenge frequently encountered in tumor cells following CAR-T cell therapy and leading to diminished antitumor efficacy and compromised long-term treatment success[78,79].

The CAR/TCR/NKR triple-targeting mechanism of [Allo15]CAR33-NKT cells was validated through a series of in vitro tumor cell killing assays using six human AML cell lines, each representing different scenarios of tumor antigen expression levels: THP1 (CD33+CD1d^high), KG1 (CD33+CD1d^medium), HL60 (CD33+CD1d^low/-), THP1^CD33-/- (CD33-CD1d+), THP1^CD1d-/- (CD33+CD1d-), and THP1^CD33/CD1d-/- (CD33-CD1d-) (Fig. 3a–c). All cell lines expressed high levels of NK ligands (i.e., CD112, CD155, and

MICA/B) and were engineered to overexpress firefly luciferase and green fluorescence protein dual reporters (FG) for monitoring via luciferase assay and flow cytometry (Fig. 3a, d). The therapeutic efficacy of three types of cells, including [Allo15]CAR33-NKT, CAR33-T, and non-CAR33-engineered PBMC-derived T (PBMC-T) cells, was assessed in these assays.

PBMC-T cells did not kill any AML tumor cells within a 24-h period (Fig. 3e and Supplementary Fig. 7a). However, when engineered with CAR, CAR33-T cells demonstrated efficient killing of all CD33+ AML tumor cells while showing no cytotoxicity towards CD33- tumor cells, demonstrating their reliance on the CAR33-CD33 interaction (Fig. 3e and Supplementary Fig. 7a). In contrast, [Allo15]CAR33-NKT cells demonstrated cytotoxicity against both CD33+ and CD33- AML tumor cells (Fig. 3e and Supplementary Fig. 7a), indicating they can target tumor cells through both CAR33-dependent and independent mechanisms. For CD1d+ AML cells (e.g., THP1-FG and THP1-FG^CD33-/-), the addition of αGC significantly enhanced antitumor efficacy of [Allo15]CAR33-NKT cells, while for CD1d- AML cells (e.g., HL60-FG and THP1-FG^CD1d-/-), the addition of αGC did not boost tumor cell killing, suggesting that [Allo15]CAR33-NKT cells can target AML tumor cells via NKT TCR-mediated pathways (Fig. 3e and Supplementary Fig. 7a). Additionally, [Allo15]CAR33-NKT cells efficiently eliminate CD33- and CD1d- AML cells (e.g., THP1-FG^CD33/CD1d-/-), and their killing capacity was diminished when NKRs (i.e., NKG2D and DNAM-1) were blocked, highlighting a reliance on an NKR-mediated targeting mechanism (Fig. 3f, g, and Supplementary Fig. 7a).

Corresponding to their multiple targeting mechanisms and strong cytotoxic capabilities (Fig. 3h), [Allo15]CAR33-NKT cells demonstrated higher effector functions compared to conventional CAR33-T cells, as evidenced by the more substantial upregulation of activation markers (i.e., CD69) and increased production of cytotoxic molecules (i.e., Perforin and Granzyme B), as well as effector cytokines (i.e., IFN-γ) (Fig. 3i, j and Supplementary Fig. 7b). Remarkably, [Allo15]CAR33-NKT cells generated through the three expansion approaches (i.e, αCD3/αCD28 Ab, αGC/PBMCs, and aAPCs) displayed comparable tumor-killing efficacy, indicating that all three expansion methods can be effectively employed to generate functional therapeutic cells for the treatment of myeloid malignancies (Supplementary Fig. 7c).

## Allogeneic CAR33-NKT cells display distinct bone marrow homing mediated by CXCR4/CCR5 expression

The BM homing capacity of therapeutic cells is essential for sustained therapeutic efficacy in the treatment of AML and MDS[13]. We assessed the PK/PD, particularly focusing on the BM homing ability of [Allo15]CAR33-NKT cells. These cells were engineered to express FG (denoted as [Allo15]CAR33-NKT/FG cells) and monitored in an NSG mouse model using bioluminescence imaging (BLI) (Fig. 4a). Conventional CAR33-T cells labeled with FG (denoted as CAR33-T/FG cells) were utilized as a benchmark control (Fig. 4a).

In vivo, the [Allo15]CAR33-NKT/FG cells rapidly expanded more than 200-fold, peaking around 30 days, followed by a gradual decline and maintaining persistence for approximately 50 days (Fig. 4b, c). Notably, [Allo15]CAR33-NKT/FG cells demonstrated strong BM homing capabilities, with the majority of these cells detected in the femur, pelvis, ribcage, and spine of the experimental mice (Fig. 4d–g). In contrast, conventional CAR33-T/FG cells showed rapid and consistent expansion over 400-fold post-injection but led to severe xenogeneic GvHD and eventual mouse mortality (Fig. 4b, c). These cells primarily infiltrated various organs, including the lungs, liver, spleen, and gastrointestinal (GI) tract, with minimal homing to the BM (Fig. 4d–g).

Compared to conventional CAR33-T/FG cells, [Allo15]CAR33-NKT/FG cells expressed higher levels of CXCR4 and CCR5 (Fig. 4h, i), key chemokine receptors that facilitate lymphocyte trafficking to the BM[80,81]. The enhanced BM homing capacity of [Allo15]CAR33-NKT/FG cells, likely driven by elevated CXCR4 and CCR5 expressions, underscores their potential therapeutic value for treating myeloid malignancies,

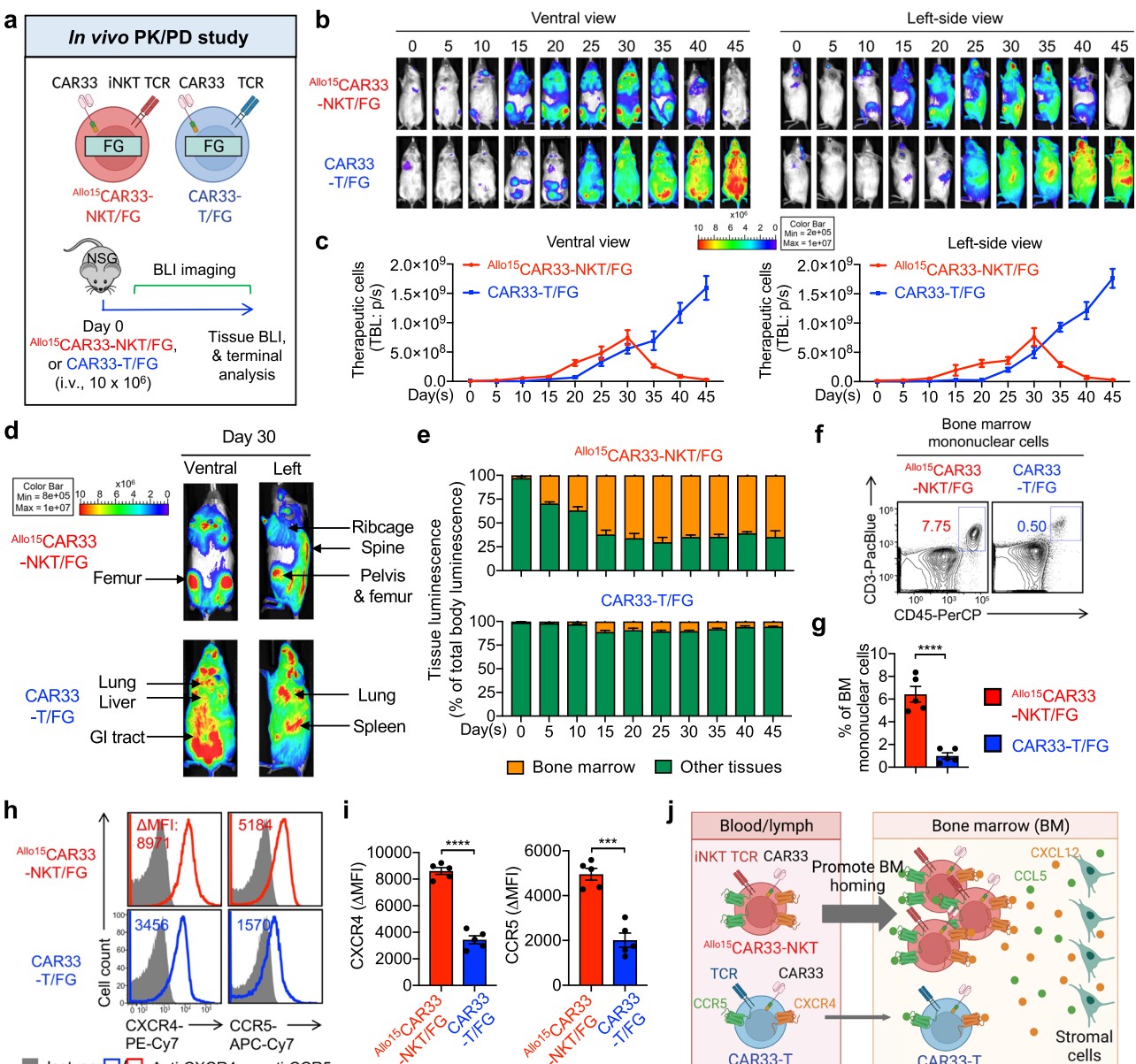

**Fig. 4 | Distinct in vivo bone marrow homing of ^Allo15^CAR33-NKT cells mediated by CXCR4/CCR5 expression. a** Experimental design to study the in vivo PK/PD of ^Allo15^CAR33-NKT cells in a xenograft NSG mouse model. The therapeutic cells were labeled with FG. Created in BioRender. LI, Y. (2025) https://BioRender.com/u95q769. **b** BLI images showing the presence of therapeutic cells in experimental mice over time. Ventral and left-side views are shown. **c** Quantification of (**b**) (*n* = 3 from three experimental mice). TBL, total body luminescence; p/s, photons per second. **d** BLI images showing the biodistribution of ^Allo15^CAR33-NKT and CAR33-T cells in representative experimental mice. Ventral and left-side views are shown. **e** Quantification of therapeutic cell tissue (i.e., bone marrow and other

tissues) distribution (*n* = 3 from three experimental mice). **f** FACS detection of therapeutic cells in the bone marrow of experimental mice 30 days after cell injection. **g** Quantification of (**f**) (*n* = 5 from five experimental mice). **h** FACS measurement of surface CXCR4 and CCR5 expressions in the indicated therapeutic cells. **i** Quantification of (**h**) (*n* = 5 from five experimental mice). **j** Schematic illustrating the distinct in vivo bone marrow homing capacity of ^Allo15^CAR33-NKT cells mediated by CXCR4/CCR5 expression. Created in BioRender. LI, Y. (2025) https://BioRender.com/g04r306. Representative of 2 experiments. Data are presented as the mean ± SEM. ***$p < 0.001$; ****$p < 0.0001$, by two-tailed Student's *t* test (**g**, **i**). Source data and exact *p* values are provided as a Source Data file.

particularly those involving refractory tumors located within the BM (Fig. 4j)[4].

## Allogeneic CAR33-NKT cells target the bone marrow-resident leukemia stem and progenitor cells in myeloid malignancies

Targeting BM-resident tumor cells, particularly LSPCs, is critical for effective therapy against AML and MDS[4]. These cells are known for their therapy resistance and immune evasion capabilities[12]. To assess the efficacy of ^Allo15^CAR33-NKT cells in targeting these BM-resident LSPCs, we utilized a series of AML and MDS xenograft mouse models alongside a cohort of primary samples from patients with AML or MDS

(Figs. 5, 6, and Supplementary Fig. 8). Five in vivo models were employed to represent different disease scenarios: the CD33+CD1d+ THP1-FG and KG1-FG xenograft mouse models (Fig. 5a–n), the CD33+CD1d^lo/-^ HL60-FG xenograft mouse model (Supplementary Fig. 8a–d), the CD33-CD1d+ THP1-FG^CD33-/-^ xenograft mouse model (Supplementary Fig. 8e–h), and an AML PDX mouse model (Fig. 6). Conventional CAR33-T cells were used as a benchmark control.

In the three human AML xenograft mouse models (THP1-FG, KG1-FG, and HL60-FG), a single administration of ^Allo15^CAR33-NKT cells successfully eliminated tumors in the majority of experimental mice (5 of 5 in both THP1-FG and KG1-FG models, and 4 of 5 in the HL60-FG

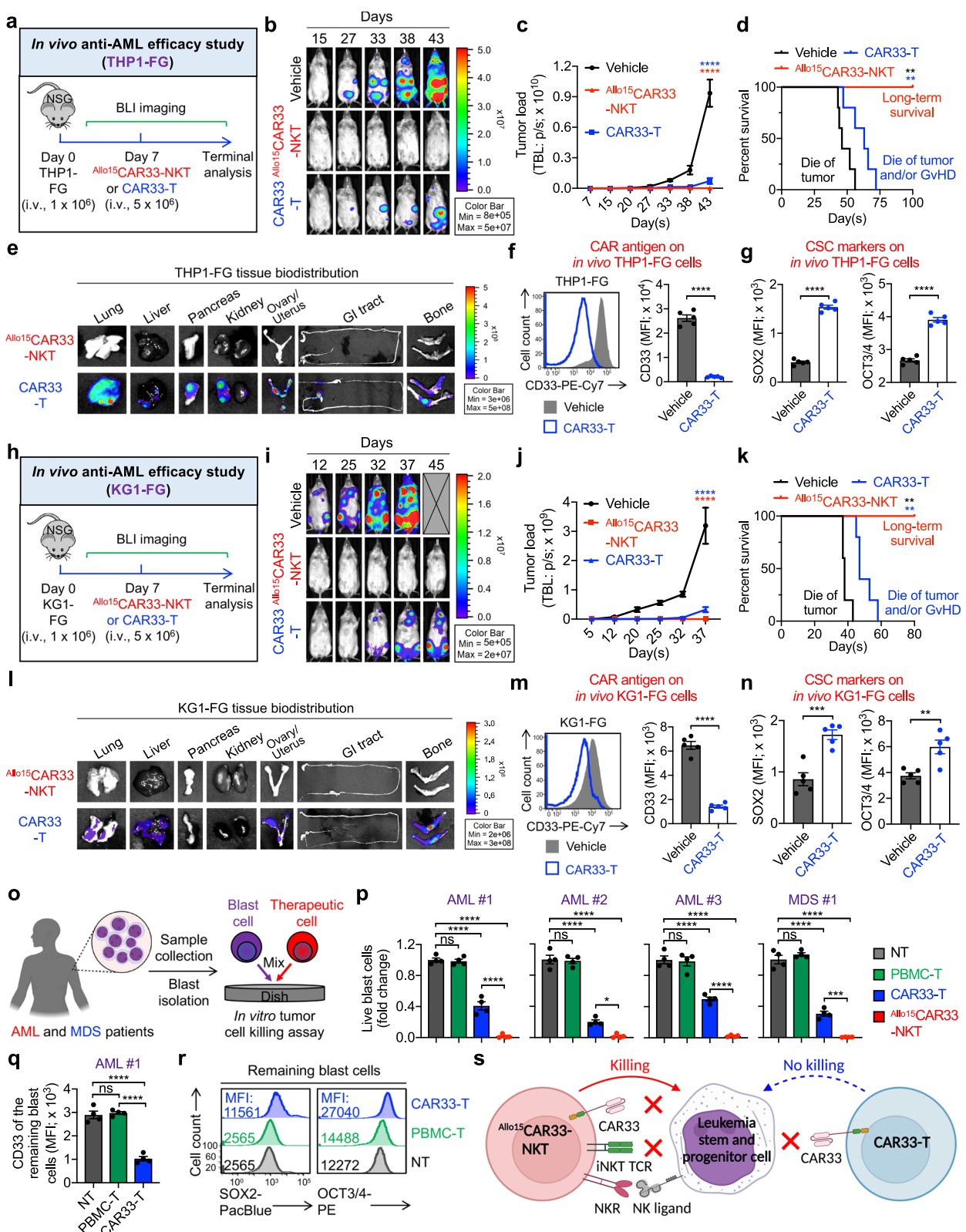

model), resulting in long-term survival of all treated mice (Fig. 5b–d, i–k, Supplementary Fig. 8b–d). In contrast, CAR33-T cells only partially suppressed tumor growth and showed limited improvement in mouse survival (Fig. 5b–d, i–k, Supplementary Fig. 8b–d). The mice treated with CAR33-T cells succumbed to a combination of tumor progression and xenogeneic GvHD (Fig. 5b–d, i–k, Supplementary Fig. 8b–d). Compared to CAR33-T cells, Allo15CAR33-NKT cells exhibited a higher

effector phenotype, as evidenced by elevated expression of activation marker (i.e., CD69) and production of cytotoxic molecules (i.e., Perforin and Granzyme B) (Supplementary Fig. 9a, b). Additionally, Allo15CAR33-NKT cells showed a reduced exhaustion phenotype, suggesting enhanced persistence and sustained activity, which may contribute to improved therapeutic outcomes in targeting AML (Supplementary Fig. 9c).

**Fig. 5 | Targeting bone marrow-resident leukemia stem and progenitor cells by [Allo15]CAR33-NKT cells using AML xenograft mouse models and primary patient samples. a**–**g** Studying the in vivo antitumor efficacy of [Allo15]CAR33-NKT cells in a THP1-FG human AML xenograft NSG mouse model. **a** Experimental design. **b** BLI images. **c** Quantification of (**b**) (n = 5). **d** Kaplan–Meier survival curves (n = 5). **e** BLI images showing the presence of residual tumor cells in mouse tissues at the termination day. GI tract, gastrointestinal tract. FACS analyses of surface CD33 (**f**) and intranuclear cancer stem cell (CSC) marker (**g**) expression in THP1-FG tumor cells, collected from mouse bone marrow at the termination day (n = 5). **h**–**n** Studying the in vivo antitumor efficacy of [Allo15]CAR33-NKT cells in a KG1-FG human xenograft NSG mouse model. **h** Experimental design. **i** BLI images. **j** Quantification of (**i**) (n = 5). **k** Kaplan–Meier survival curves (n = 5). **l** BLI images showing the presence of residual tumor cells in mouse tissues at the termination day. FACS analyses of surface CD33 (**m**) and intranuclear CSC marker (**n**) expression in KG1-FG tumor cells collected from mouse bone marrow at the termination day (n = 5). **o**–**r** Studying the

antitumor efficacy of [Allo15]CAR33-NKT cells against primary patient samples. **o** Experimental design. Created in BioRender. LI, Y. (2025) https://BioRender.com/o37h997 **p** Blast cell killing data at 24 h (n = 4 from four different therapeutic cell donors). Data from three AML and one MDS patient samples are presented. FACS analyses of surface CD33 (**q**) and intranuclear CSC marker **r** expression in the remaining blast cells collected after the 24-h in vitro tumor cell killing assay (n = 4 from four different therapeutic cell donors). **s** Diagram illustrating the ability of [Allo15]CAR33-NKT cells to target CD33-low/negative LSPCs in myeloid malignancies. Created in BioRender. LI, Y. (2025) https://BioRender.com/m06k581. Representative of 2 (**a**–**n**) and 3 (**o**–**r**) experiments. Data are presented as the mean ± SEM. ns, not significant; *p < 0.05; **p < 0.01; ***p < 0.001; ****p < 0.0001, by two-tailed Student's t test (**f**, **g**, **m**, **n**), one-way ANOVA (**c**, **j**, **p**, **q**), or log rank (Mantel-Cox) text adjusted for multiple comparisons (**d**, **k**). Source data and exact p values are provided as a Source Data file.

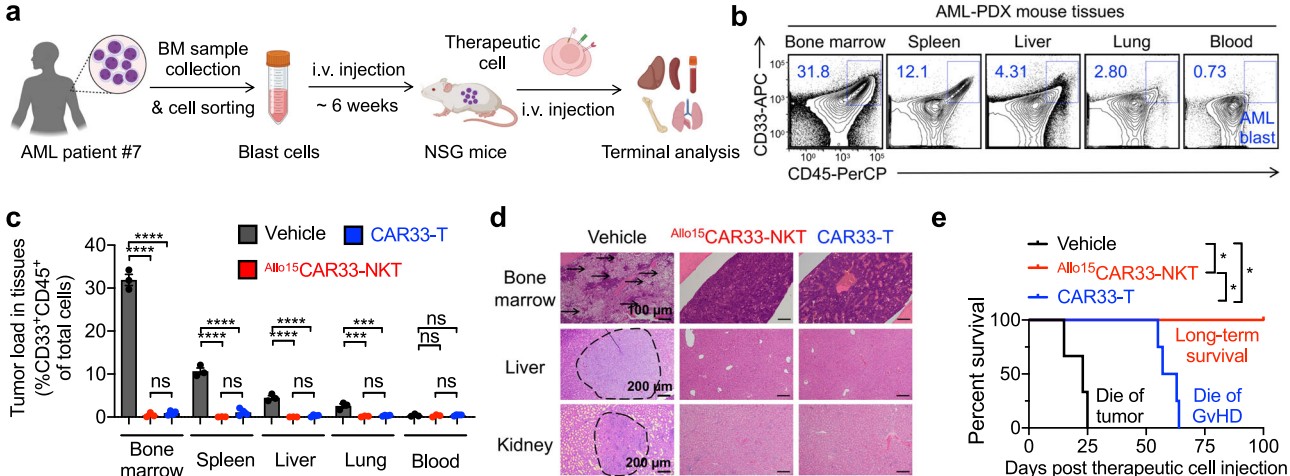

**Fig. 6 | In vivo tumor targeting efficacy of [Allo15]CAR33-NKT cells in a human AML patient-derived xenograft (PDX) NSG mouse model. a** Experimental design. Created in BioRender. LI, Y. (2025) https://BioRender.com/o37h997. **b** FACS detection of AML blast cells (gated as CD33+CD45+ cells) in various tissues collected from experimental mice at the termination day. **c** FACS analyses showing the AML blast cell loads in various tissues of experimental mice (n = 3 for Vehicle and [Allo15]CAR33-NKT; n = 4 for CAR33-T). Tissues from both the Vehicle and CAR33-T cell groups were collected from experimental mice at the termination day. Tissues from the [Allo15]CAR33-NKT cell group were collected from experimental mice at day 100. **d** H&E-stained tissue sections. Tissues were collected from experimental mice at day 20 post therapeutic cell injection. Scale bars, 100 μm for bone marrow samples, 200 μm for liver and kidney samples. Three independent tissue samples from each group were analyzed, and one representative data are presented. **e** Kaplan–Meier survival curves of experimental mice over time (n = 3 for Vehicle and [Allo15]CAR33-NKT; n = 4 for CAR33-T). Representative of 2 experiments. Data are presented as the mean ± SEM. ns not significant, *p < 0.05; **p < 0.01; ***p < 0.001; ****p < 0.0001, by one-way ANOVA (**c**), or log rank (Mantel-Cox) text adjusted for multiple comparisons (**e**). Source data and exact p values are provided as a Source Data file.

Tissue analyses revealed that following treatment with CAR33-T cells, AML tumor cells relapsed in various organs, including the BM, lungs, and liver (Fig. 5e, l). In contrast, treatment with [Allo15]CAR33-NKT cells resulted in the complete absence of tumors across all examined tissues, including the BM (Fig. 5e, l). Notably, compared to untreated AML tumor cells, those that relapsed after CAR33-T cell therapy demonstrated a downregulation of CD33 and an upregulation of CSC markers, such as SOX2 and OCT3/4 (Fig. 5f, g, m, n)[82]. This suggests that the relapsed AML tumor cells may have undergone CAR antigen escape and acquired enhanced CSC characteristics, potentially contributing to their ability to evade CAR-T cell therapy and eventually leading to disease relapse.

Indeed, in the THP1-FG[CD33-/-] human AML xenograft mouse model which was designed to simulate the scenario of tumor antigen escape (Supplementary Fig. 8e), a single administration of [Allo15]CAR33-NKT cells effectively suppressed tumor growth and led to prolonged survival of the experimental mice (Supplementary Fig. 8f–h). Conversely, CAR33-T cells exhibited limited efficacy in tumor suppression, possibly attributable to the graft-versus-leukemia (GvL) effect[83], and did not confer improvement in mouse survival (Supplementary Fig. 8f–h). These findings underscore that even in conditions where conventional CAR-T cells offer limited benefits due to tumor antigen escape,

[Allo15]CAR33-NKT cells retain potent antitumor capacity, likely owing to their utilization of multiple tumor-targeting mechanisms, including NKR and TCR-mediated killing (Fig. 3h).

In addition, we established an AML PDX model to assess the in vivo therapeutic efficacy of [Allo15]CAR33-NKT cells (Fig. 6a). In a high-risk AML disease context, we achieved significant engraftment of malignant blast cells in various mouse organs, including BM, liver, and kidney, with these AML cells demonstrating high CD33 expression levels (Fig. 6b–d). Flow cytometry and tissue histology analyses demonstrated the successful elimination of malignant blast cells by both CAR33-T and [Allo15]CAR33-NKT cells (Fig. 6c, d). However, only [Allo15]CAR33-NKT cells conferred long-term mouse survival, while CAR33-T cells ultimately resulted in mouse mortality due to xenogeneic GvHD (Fig. 6e). Collectively, [Allo15]CAR33-NKT cells exhibit potent antitumor efficacy against AML in vivo across multiple models, highlighting their promising therapeutic potential in treating these myeloid malignancies.

We subsequently verified the targeting of BM-derived LSPCs by [Allo15]CAR33-NKT cells using primary AML and MDS patient samples (Fig. 5o). In these samples, CMPs and MBCs exhibited high CD33 expression, whereas LSPCs displayed low CD33 levels, potentially evading CAR33-T cell targeting (Fig. 1c–e). Indeed, in vitro tumor cell

killing assays demonstrated that [Allo15]CAR33-NKT cells were markedly more effective at eliminating primary blast cells compared to CAR33-T cells, notably targeting CD33-negative LSPCs, which CAR33-T cells failed to engage (Fig. 5p). Post-treatment analyses showed that compared to non-treated tumor cells, the tumor cells remaining after CAR33-T cell treatment were CD33-negative/low and exhibited elevated levels of CSC markers such as SOX2 and OCT3/4 (Fig. 5q, r). In conclusion, [Allo15]CAR33-NKT cells possess a significant capacity to target bone marrow-resident LSPCs that downregulate CD33 and upregulate CSC markers (Fig. 5s). These characteristics enable [Allo15]CAR33-NKT cells to effectively address key challenges associated with tumor refractoriness and resistance to therapy in myeloid malignancies.

## Allogeneic CAR33-NKT cells display a gene profile associated with mixed T/NK cell features, strong effector/memory function and reduced exhaustion

To delve into the genomic and molecular traits of [Allo15]CAR33-NKT cells following tumor challenge, we established an in vitro tumor rechallenge assay (Supplementary Fig. 10a). This involved successive 10-day tumor rechallenges, after which live [Allo15]CAR33-NKT cells were collected for scRNA-seq. The [Allo15]CAR33-NKT cells without tumor rechallenge, along with conventional CAR33-T cells both with and without tumor rechallenge, were included as controls (Supplementary Figs. 10 and 11).

Uniform manifold approximation and projection (UMAP) analysis of combined four samples revealed the formation of five major cell clusters (Supplementary Fig. 10b, 11). The signature gene profiling and Gene Set Enrichment Analysis (GSEA) identified cluster 1 cells as proliferating cells, cluster 2 cells as effector cells, cluster 3 cells as memory cells, cluster 4 cells as exhausted cells and cluster 5 cells as resting cells (Supplementary Fig. 10b and 11)[44,84–86].

[Allo15]CAR33-NKT cells, both with and without tumor rechallenge, exhibited a mixed T/NK cell phenotype (Supplementary Fig. 10c). Compared to CAR33-T cells, [Allo15]CAR33-NKT cells displayed fewer T cell characteristics and significantly enhanced NK cell features (Supplementary Fig. 10d), aligning with their NKT cell nature, high expression of NKRs, and elevated NK-related cytotoxicity (Figs. 2i–k and 3g, h).

UMAP analysis of individual samples revealed distinct behaviors and molecular features between [Allo15]CAR33-NKT and CAR33-T cells. [Allo15]CAR33-NKT cells with tumor rechallenge displayed comparable cell populations to those without tumor rechallenge, albeit with a slightly reduced proliferating population (Cluster 1) (Supplementary Fig. 10e). In contrast, CAR33-T cells, after repeated tumor challenge, showed a significant reduction in proliferating population (Cluster 1), a comparable effector population (Cluster 2), and an increase in memory and exhausted populations (Clusters 3 and 4) (Supplementary Fig. 10e). These findings suggest that [Allo15]CAR33-NKT cells maintain better functionality post tumor response, likely due to their IL-15-enhanced persistence and long-term efficacy[44,60,87].

We conducted detailed analyses of two therapeutic cell samples post repeated tumor challenges. Compared to CAR33-T cells, [Allo15]CAR33-NKT cells exhibited heightened proliferating and effector characteristics, comparable memory feature, and significantly reduced exhaustion property (Supplementary Fig. 10f, g). Consistent with the in vivo flow cytometry results (Supplementary Fig. 9), following tumor rechallenge, [Allo15]CAR33-NKT cells maintained high expression levels of genes encoding effector molecules and cytotoxic molecules (e.g., *GZMB*, *PRF1*, *KLRC1*, and *KLRC2*), and showed low expression of genes encoding exhaustion markers (e.g., *LAG3*, *CTLA4*, and *TIGIT*) (Supplementary Fig. 10g). The robust effector/memory function and reduced exhaustion of [Allo15]CAR33-NKT cells confer a potent ability to target myeloid malignancies with sustained efficacy (Figs. 5a–n, 6, and Supplementary Fig. 8). Supporting this, Pesudotime and CytoTRACE analyses revealed the trajectory of therapeutic cells, indicating accumulation of [Allo15]CAR33-NKT cells at the proliferating and/or effector stage, while conventional CAR33-T cells were observed to remain at the memory and/or exhausted stage (Supplementary Fig. 10h–j).

We then investigated the gene expression changes in therapeutic cell samples post-tumor treatment compared to pre-tumor treatment. Following tumor cell treatment, [Allo15]CAR33-NKT cells showed upregulation of genes associated with positive regulation of metabolic processes and cellular response to stress (Supplementary Fig. 10k). Conversely, CAR33-T cells upregulated genes related to negative regulation of biosynthesis and metabolic processes (Supplementary Fig. 10k). Importantly, [Allo15]CAR33-NKT cells exhibited a higher capacity for metabolism, including glycolysis, oxidative phosphorylation, and electron transport chain activities, compared to CAR33-T cells (Supplementary Fig. 10l). In addition, [Allo15]CAR33-NKT cells demonstrated a comparable level of fatty acid metabolism (Supplementary Fig. 10l). The distinct metabolic profile of [Allo15]CAR33-NKT cells during antitumor responses suggests their potential to target tumor cells more effectively, particularly in nutrient-deficient tumor microenvironments[88].

## Allogeneic CAR33-NKT cells can synergize with hypomethylating agent treatment

HMAs, including Azacitidine and Decitabine, have emerged as cornerstone therapies in the standard clinical management of AML and MDS, offering significant therapeutic benefits by modulating aberrant DNA methylation patterns and thereby restoring normal gene expression profiles[10]. They have demonstrated the ability to induce the upregulation of various markers on tumor cells. These include tumor suppressor genes (e.g., p53), differentiation markers (e.g., CD14, CD11b, and CD70), inhibitory immune checkpoint ligands (e.g., PD-L1 and PD-L2), NK ligands (e.g., MICA and ULBP), and CD1d[10,24,48,89]. The upregulation of NK ligands and CD1d following HMA treatment suggests potential synergism with [Allo15]CAR33-NKT cell therapy, potentially leading to enhanced efficacy against AML and MDS.

The human AML cell line THP1 was employed to investigate the effects of HMAs on antigen expression and susceptibility to [Allo15]CAR33-NKT cells (Fig. 7a). Flow cytometry analyses revealed that following repeated Decitabine treatment, THP1 cells exhibited a significant increase in the expression of NK ligands (i.e., CD112, MICA/B, and ULBP) as well as CD1d (Fig. 7b, c). Notably, there was no observed upregulation of CD33 following HMA treatment (Fig. 7b, c). Consequently, HMAs may not enhance the tumor cell killing ability of conventional CAR33-T cells, as these cells rely solely on CAR-mediated killing mechanisms (Fig. 3h). However, HMAs are likely to facilitate the tumor cell killing activity of [Allo15]CAR33-NKT cells by augmenting their NKR- and TCR-mediated cytotoxicity against tumor cells (Fig. 3h).

In an in vitro tumor cell killing assay, conventional CAR33-T cells exhibited comparable antitumor efficacy to THP1 cells, regardless of Decitabine treatment (Fig. 7d). Conversely, [Allo15]CAR33-NKT cells demonstrated enhanced cytotoxicity against Decitabine-treated THP1 cells (Fig. 7d). Furthermore, in the presence of αGC supplementation, which activates NKT TCR-mediated killing, [Allo15]CAR33-NKT cells also exhibited augmented killing of Decitabine-treated THP1 cells (Fig. 7d). Consistently, [Allo15]CAR33-NKT cells displayed increased production of cytotoxic molecules, such as Granzyme B, when targeting Decitabine-treated tumor cells (Fig. 7e, f). This heightened cytotoxic response suggests that Decitabine-treated tumor cells are more susceptible to [Allo15]CAR33-NKT cell-mediated killing, likely due to the upregulation of NKR ligands and CD1d on the tumor cells (Fig. 7g).

Subsequently, we investigated the synergistic potential of [Allo15]CAR33-NKT cells and HMAs using the THP1-FG and KG1-FG human AML xenograft NSG mouse models (Fig. 7h and Supplementary Fig. 12a). While both low doses of [Allo15]CAR33-NKT cells and Decitabine treatment individually achieved tumor

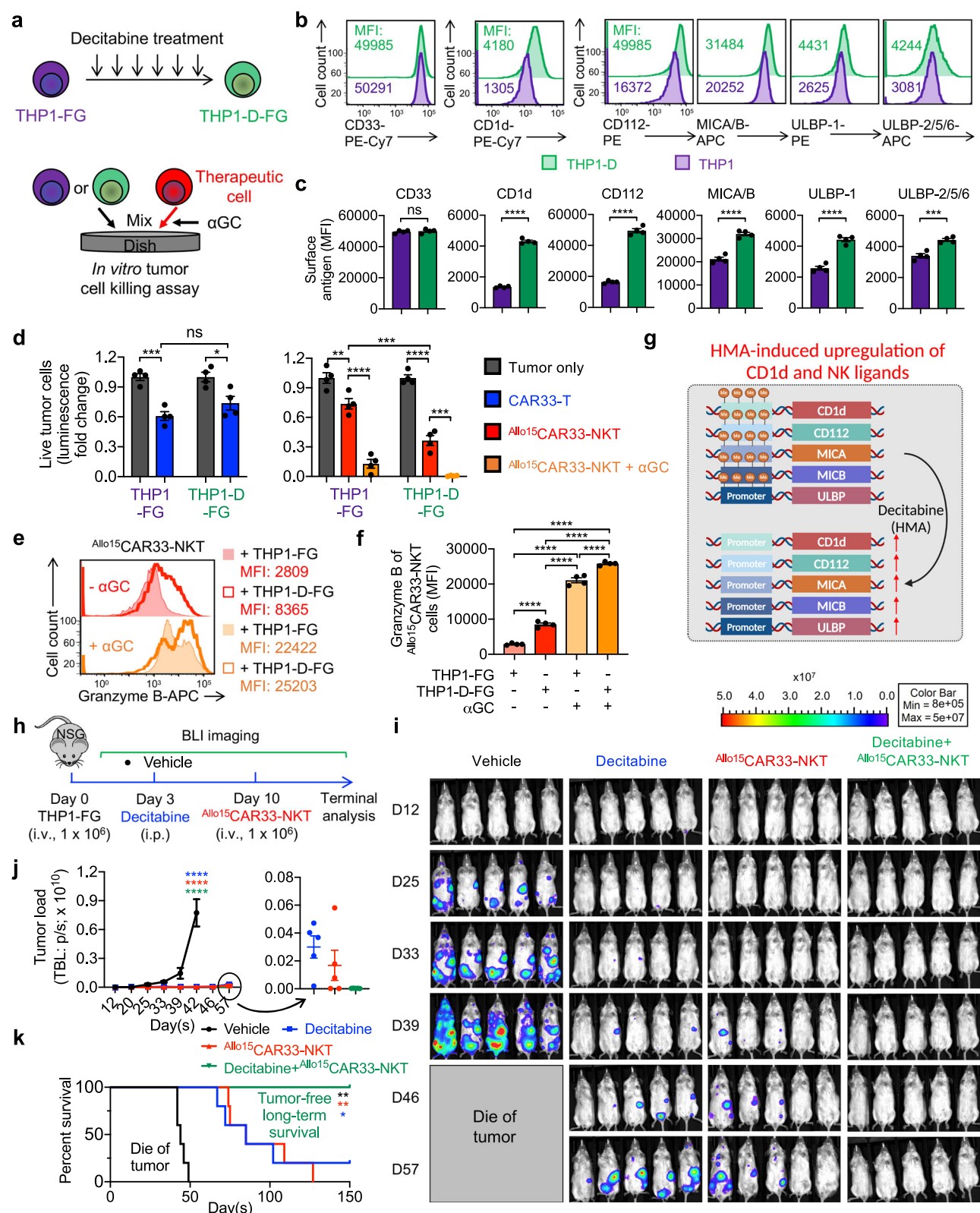

suppression, the combination of these two therapies resulted in the most favorable therapeutic outcomes, leading to complete tumor elimination and long-term survival in all mice (Fig. 7i–k and Supplementary Fig. 12b, c). These findings underscore the synergistic efficacy of combining $^{Allo15}$CAR33-NKT cell therapy with HMAs in the treatment of AML. This combinatorial approach holds promise for improving treatment outcomes and warrants further investigation in clinical settings.

## Allogeneic CAR33-NKT cells exhibit a high safety profile characterized by minimal on-target off-tumor effect against hematopoietic precursors

The expression of CD33 on various hematopoietic precursor cells, such as myeloid progenitor cells, monocytes, macrophages, granulocytes, and dendritic cells (DCs), raises concerns regarding the potential for on-target off-tumor effects[90]. Therefore, we aimed to investigate the on-target off-tumor effect of $^{Allo15}$CAR33-NKT cells against

**Fig. 7 | Synergistic effect of $^{Allo15}$CAR33-NKT cells with hypomethylating agent (HMA) in the treatment of myeloid malignancies. a–f** Studying the synergistic effect of $^{Allo15}$CAR33-NKT cells with HMA Decitabine using an in vitro tumor cell killing assay. **a** Experiment design. THP1-D, THP1 tumor cells treated with decitabine. **b** FACS detection of CAR target (CD33), iNKT TCR target (CD1d), and NKR targets (i.e., CD112, MICA/B, ULBP-1, and ULBP-2/5/6) on the indicated AML cells. **c** Quantification of (**b**) ($n = 4$ from four different experimental batches). **d** Tumor cell killing data at 24 h ($n = 4$ from four different experimental batches). **e** FACS detection of intracellular Granzyme B production by $^{Allo15}$CAR33-NKT cells. **f** Quantification of (**e**) ($n = 4$ from four different experimental batches). **g** Diagram showing the upregulation of CD1d and NK ligands on AML tumor cells following

treatment with HMA. Created in BioRender. FANG, Y. (2025) https://BioRender.com/n85n160. **h–k** Studying the in vivo synergistic effect of $^{Allo15}$CAR33-NKT cells with HMA Decitabine using a THP1-FG human AML xenograft NSG mouse model. **h** Experimental design. **i** BLI images showing the presence of tumor cells in experimental mice over time. **j** Quantification of (**i**) ($n = 5$). **k** Kaplan–Meier survival curves of experimental mice over time ($n = 5$). Representative of 2 (**h–k**) and 3 (**a–g**) experiments. Data are presented as the mean ± SEM. ns not significant, *$p < 0.05$; **$p < 0.01$; ***$p < 0.001$; ****$p < 0.0001$, by two-tailed Student's $t$ test (**c**), one-way ANOVA (**d, f, j**), or log rank (Mantel-Cox) text adjusted for multiple comparisons (**k**). Source data and exact $p$ values are provided as a Source Data file.

hematopoietic precursors using in vitro assays including the HSPC killing assay (Fig. 8a) and the HSPC colony formation assay (Fig. 8f), as well as in vivo experimentation utilizing a BLT (human bone marrow-liver-thymus engrafted NOD/SCID/γc$^{-/-}$ mice) humanized mouse model (Fig. 8i).

Initially, we assessed CD33 expression across various immune cell subsets derived from healthy donors. Only monocytes and granulocytes exhibited elevated levels of CD33, whereas other cell populations, including CD34$^+$ HSPCs, lacked CD33 expression (Fig. 8b). Analyses of CD33 mRNA expression across various human tissue cells further corroborated its specific expression within myeloid lineage cells (Supplementary Fig. 13a). Consequently, CD33$^+$ monocytes and granulocytes are susceptible to targeting by CAR33-T and $^{Allo15}$CAR33-NKT cells, while CD33$^-$ HSPCs and other immune cells are not susceptible to these therapeutic cells (Fig. 8c). Furthermore, HSPCs were found to lack expression of CD1d (Fig. 8d), indicating that they are not susceptible to targeting by $^{Allo15}$CAR33-NKT cells, even in the presence of glycolipid antigens (Fig. 8e).

We then employed an in vitro HSPC colony assay to assess the impact of $^{Allo15}$CAR33-NKT cells on hematopoietic precursors (Fig. 8f). CD34$^+$ HSPCs obtained from healthy donors were exposed to $^{Allo15}$CAR33-NKT and CAR33-T cells for 4 h before being allowed to develop into colonies (Fig. 8f)[22]. Importantly, neither $^{Allo15}$CAR33-NKT nor CAR33-T cells exhibited any effect on the formation of Colony-Forming Unit-Granulocyte/Macrophage (CFU-GM) and Burst-Forming Unit-Erythroid (BFU-E) colonies, indicating the safety of utilizing both therapeutic cell products (Fig. 8g, h).

To further evaluate the safety of $^{Allo15}$CAR33-NKT cells on hematopoietic precursors in vivo, we established a BLT humanized mouse model (Fig. 8i). In this model, human CD34$^+$ HSPCs were inoculated into NSG mice, and human thymus tissue was implanted to support the differentiation of CD34$^+$ HSPCs into various lineages, including T, B, NK, and myeloid cells (Fig. 8j and Supplementary Fig. 13b–d)[91,92]. Among these cells, only myeloid lineage cells expressed CD33, while other cell types, including HSPCs, T cells, B cells, and NK cells, did not express CD33 (Fig. 8k and Supplementary Fig. 13e). At 8 weeks post-HSPC inoculation, $^{Allo15}$CAR33-NKT and CAR33-T cells were injected into these BLT mice to assess their targeting of hematopoietic precursors (Fig. 8i).

Both $^{Allo15}$CAR33-NKT and CAR33-T cells exhibited a high safety profile, as evidenced by body weight and clinical score analyses (Fig. 8l, m). However, CAR33-T cells induced mild body weight loss and symptoms such as mild hunched posture and dehydration during the first week post-injection (Fig. 8i, m). Terminal analyses demonstrated that both $^{Allo15}$CAR33-NKT and CAR33-T cells effectively killed CD33$^+$ cells such as myeloid cells and DCs, while sparing CD33$^-$ cells such as HSPCs, T cells, B cells, and NK cells in various organs, including the bone marrow, spleen, liver, and peripheral blood (Fig. 8n–p and Supplementary Fig. 13f). Notably, some CD33$^+$ myeloid cells and DCs in the liver were not completely eliminated (Fig. 8p), suggesting that both $^{Allo15}$CAR33-NKT and CAR33-T cells might preserve certain tissue-resident myeloid cells. Collectively, these results suggest that $^{Allo15}$CAR33-NKT cells demonstrate a high safety profile, characterized by minimal on-target off-tumor effects against hematopoietic precursors.

## Allogeneic CAR33-NKT cells exhibit a high safety profile characterized by minimal GvHD risk and low CRS attributes

One of the primary safety considerations with allogeneic cell therapy involves the potential risk of GvHD, wherein donor immune cells attack the recipient's tissues[93,94]. $^{Allo15}$CAR33-NKT cells present a distinct advantage in this regard due to their unique NKT TCR recognition of the nonpolymorphic MHC molecule CD1d, suggesting a reduced likelihood of inducing GvHD compared to allogeneic conventional αβ T cell therapies[95–98]. This characteristic was assessed through an in vitro mixed lymphocyte reaction (MLR) assay (Fig. 9a, b) and an in vivo xenograft NSG mouse model (Fig. 9c–h).

In the in vitro MLR assay, $^{Allo15}$CAR33-NKT cells were stimulated with irradiated CD33-negative PBMCs derived from a diverse set of mismatched healthy donors (>10 donors) (Fig. 9a). Notably, the PBMCs were pre-sorted to deplete CD33-positive cells, eliminating the potential for on-target activity mediated by CAR33. The assay demonstrated minimal production of IFN-γ by $^{Allo15}$CAR33-NKT cells, in stark contrast to conventional CAR33-T cells, which exhibited robust IFN-γ production (Fig. 9b). This suggests that $^{Allo15}$CAR33-NKT cells pose a lower risk of inducing a GvH response.

In the in vivo xenograft NSG mouse model, the administration of conventional CAR33-T cells resulted in significant expansion of conventional T cells within experimental mice after one month (Fig. 4b, c), accompanied by the development of severe xenogeneic GvHD characterized by noticeable reductions in body weight and clinical symptoms of GvHD, ultimately leading to the mortality of experimental mice (Fig. 9d–f). The induction of xenogeneic GvHD was further evidenced by substantial infiltration of immune cells into vital organs, such as the liver and lung (Fig. 9g, h). In stark contrast, treatment with $^{Allo15}$CAR33-NKT cells conferred a GvHD-free, long-term survival outcome for all experimental mice (Fig. 9d–f), associated with the absence of immune cell infiltration into vital organs (Fig. 9g, h).

Cytokine release syndrome (CRS) represents a significant concern in CAR-T cell therapy due to its potential for severe adverse effects[99–101]. Notably, studies employing human tumor xenograft mouse models have revealed a contributory role of mouse macrophages in exacerbating CRS effects[99,102]. Intriguingly, in a murine model bearing heavy tumor burden, treatment with $^{Allo15}$CAR33-NKT cells demonstrated a distinct advantage over CAR33-T cells, manifesting in less changes of body weight and reduced levels of CRS-associated biomarkers, such as mouse IL-6 and serum amyloid A-3 (SAA-3), in both mouse serum and peritoneal fluid (Fig. 9i–l). These findings suggest that $^{Allo15}$CAR33-NKT cells may offer a reduced risk of CRS-like responses, potentially attributed to their NK cell characteristics (Figs. 2i, j and 3f, g) and their capability to mitigate macrophage-mediated CRS exacerbation[46,103].

Remarkably, $^{Allo15}$CAR33-NKT cells exhibited enduring safety, as evidenced by minimal organ damage observed 120 days following adoptive transfer into NSG mice (Fig. 9m). These findings collectively underscore a promising safety profile for $^{Allo15}$CAR33-NKT cells, thereby bolstering their potential for off-the-shelf therapeutic applications.

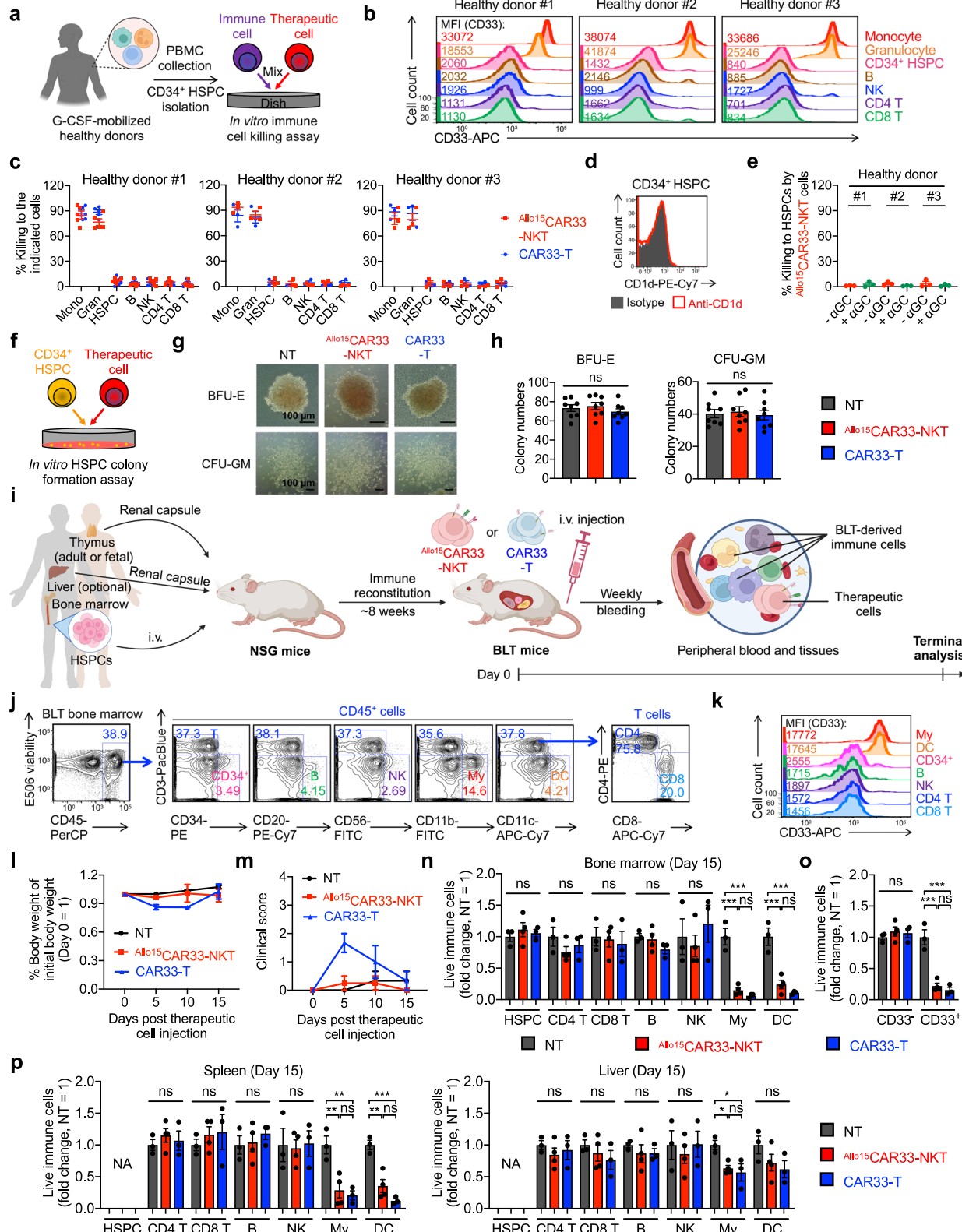

## Allogeneic CAR33-NKT cells resist host cell-mediated allorejection

To evaluate host cell-mediated allorejection against [Allo15]CAR33-NKT cells, we performed two in vitro MLR assays (Supplementary Fig. 14a, c). The first MLR assay was designed to study T cell-mediated allorejection, wherein irradiated [Allo15]CAR33-NKT cells were cocultured with donor-mismatched PBMCs, followed by measuring IFN-γ production by the PBMCs (Supplementary Fig. 14a). Compared to CAR33-T cells, [Allo15]CAR33-NKT cells induced less IFN-γ production, indicating their resistance to T cell-mediated allorejection (Supplementary Fig. 14b). The second MLR assay was designed to study NK cell-mediated allorejection, wherein [Allo15]CAR33-NKT cells were cocultured with donor-mismatched PBMC-derived NK (PBMC-NK) cells, followed by detecting viable [Allo15]CAR33-NKT cells (Supplementary Fig. 14c). Compared to CAR33-T cells,

**Fig. 8 | On-target off-tumor effect of** $^{Allo15}$**CAR33-NKT cells against hematopoietic precursors. a–e** Studying the HSPC targeting using an in vitro immune cell killing assay. **a** Experimental design. G-CSF, granulocyte colony-stimulating factor. Created in BioRender. LI, Y. (2025) https://BioRender.com/o37h997 **b** FACS detection of CD33 expression on the indicated immune cells. **c** Immune cell killing data at 24 h ($n = 3$ for healthy donor 1, and $n = 5$ for healthy donors 2 and 3; n indicates different therapeutic cell batches). **d** Flow detection of CD1d expression on HSPCs. **e** HSPC killing data at 24 h ($n = 3$; n indicates different therapeutic cell batches). **f–h** Studying the hematopoietic precursor targeting using an in vitro HSPC colony formation assay. **f** Experimental design. **g** Images showing the formation of Burst-Forming Unit-Erythroid (BFU-E) and Colony-Forming Unit-Granulocyte/Macrophage (CFU-GM) colonies. **h** Quantification of (**g**) ($n = 8$). **i–p** Studying the hematopoietic precursor targeting using an in vivo bone marrow-liver-thymus (BLT) humanized mouse model. **i** Experimental design. Created in BioRender. FANG, Y. (2025) https://BioRender.com/g87v886 **j** FACS detection of human immune cells in the bone marrow collected from BLT mice 8 weeks post HSPC

injection and prior to therapeutic cell injection. My, myeloid cell; DC, dendritic cell. **k** FACS detection of CD33 expression on the indicated immune cells. For (**j**, **k**) data from three independent mice were analyzed, and one representative data are presented. **l** Body weight measured over time. **m** Clinical scores recorded over time. The score was calculated as the sum of individual scores of 5 categories (activity, posture, dehydration, diarrhea, and dishevelment; score 0–1 for each category). **n** FACS analyses of immune cell targeting in bone marrow on Day 15. The percentage of the indicated immune cells among total CD45$^+$CAR$^-$ immune cells from each experimental mouse was recorded, and the fold change was calculated by normalizing to the NT group. FACS analyses of CD33$^+$ and CD33$^-$ cell targeting in bone marrow (**o**), and immune cell targeting in spleen and liver (**p**). NA not available. In (**l–p**), $n = 3$ for NT and CAR33-T, and $n = 4$ for $^{Allo15}$CAR33-NKT; n indicates different experimental mice. Representative of 2 (**i–p**) and 3 (**a–h**) experiments. Data are presented as the mean ± SEM. ns not significant, *$p < 0.05$; **$p < 0.01$; ***$p < 0.001$; ****$p < 0.0001$, by one-way ANOVA (**h**, **n**, **o**, **p**). Source data and exact $p$ values are provided as a Source Data file.

$^{Allo15}$CAR33-NKT cells showed greatly improved survival, indicating their resistance to NK cell-mediated allorejection (Supplementary Fig. 14d).

To elucidate the mechanisms underlying the allorejection resistance of $^{Allo15}$CAR33-NKT cells, our initial investigations focused on cell surface molecules implicated in T cell and NK cell-mediated allorejection. Compared to CAR33-T cells, $^{Allo15}$CAR33-NKT cells exhibited reduced expression of surface HLA-I, HLA-II, and NK cell ligands (e.g., ULBP-1, MICA/B) across both in vitro cultures and in vivo antitumor assays (Supplementary Fig. 14e–i). Additionally, scRNA-seq analyses confirmed persistently low expression of gene sets encoding for HLA class I/II molecules and NK cell ligands (Supplementary Fig. 14h). These results suggest that $^{Allo15}$CAR33-NKT cells possess an intrinsic and stable hypoimmunogenic phenotype.

### Allogeneic CAR33-NKT cells outperform IL-15-enhanced conventional CAR33-T ($^{15}$CAR33-T) cells

IL-15 has been incorporated into both autologous CAR-NKT and CAR-T cell therapies in clinical trials, demonstrating enhanced efficacy in both cell types[60,104]. Notably, IL-15 did not exhibit toxicity in CAR-NKT cell products, whereas it was associated with increased CRS toxicity in cancer patients treated with conventional CAR-T cells[60,104]. Therefore, we evaluated the efficacy and safety of $^{Allo15}$CAR33-NKT cells in comparison to healthy donor PBMC-derived IL-15-enhanced conventional CAR33-T ($^{15}$CAR33-T) cells (Fig. 10a–d).

In an in vitro tumor cell killing assay, three therapeutic cell types were evaluated, including $^{Allo15}$CAR33-NKT cells, CAR33-T cells, and $^{15}$CAR33-T cells (Fig. 10e). All three therapeutic cell types exhibited significant tumor cell killing capacity; however, $^{Allo15}$CAR33-NKT cells outperformed both conventional CAR33-T and $^{15}$CAR33-T cells (Fig. 10f). This enhanced effectiveness may be attributed to their superior cytotoxicity and multiple tumor cell-targeting mechanisms (Fig. 3).

Subsequently, we investigated the in vivo antitumor efficacy and safety of the three therapeutic cell types using both THP1-FG and KG1-FG human AML xenograft NSG mouse models. In both models, all three therapeutic cells were able to suppress tumor growth; however, CAR33-T cells exhibited the lowest antitumor capacity (Fig. 10g–n). $^{Allo15}$CAR33-NKT cells showed comparable or superior antitumor efficacy compared to $^{15}$CAR33-T cells (Fig. 10g–n). Notably, mice treated with $^{15}$CAR33-T cells experienced CRS and xenogeneic GvHD side effects, ultimately leading to mortality (Supplementary Fig. 15). This finding highlights that IL-15 contributes significantly to toxicity in conventional CAR-T cell therapy, consistent with a recent clinical trial involving IL-15-enhanced GPC3-targeting CAR-T cells for the treatment of hepatocellular carcinoma (HCC) patients[104]. In contrast, only the mice treated with $^{Allo15}$CAR33-NKT cells achieved tumor-free long-term survival (Fig. 10g–n), indicating that $^{Allo15}$CAR33-NKT cells outperform $^{15}$CAR33-T cells in the treatment of myeloid malignancies in terms of both efficacy and safety.

## Discussion

Currently, conventional CAR-T cell therapy targeting tumor antigens such as CD33[30–33], CD123[105], CD44v6[106], and CLL1[107] has demonstrated efficacy and safety in treating myeloid malignancies in clinical trials. These therapies have achieved complete and long-term remissions in patients with refractory/relapsed AML and high-risk MDS, accompanied by notable levels of myeloblast ablation[34]. Although adverse events such as CRS and immune effector cell-associated neurotoxicity syndrome (ICANS) are a consistent concern in conventional CAR-T cell therapy, they typically remain manageable in clinical trials, with the majority of cases falling below grade 3 severity[33]. Additionally, a clinical trial investigating the use of CD33-directed CAR-T cells in patients with relapsed and refractory AML has reported promising outcomes in managing febrile syndromes, with significant fever reduction observed following a 12-hour administration of anti-TNF-α treatment[30]. Despite these encouraging outcomes, financial and time challenges still persist in the manufacturing of autologous CAR-T cell therapies. While allogeneic CAR-T therapy holds potential to address these challenges, it introduces other complexities, such as labor-intensive genetic engineering strategies involving TCR ablation to prevent GvHD, HLA-I/II ablation to mitigate host cell-mediated allorejection, and CD52 ablation to avoid lymphoid depletion using anti-CD52 antibodies[38,43,108,109]. These methodologies, while promising, are accompanied by inherent hurdles, including suboptimal gene editing efficiency, off-target mutations, and potential compromise of TCR functionality critical for T cell longevity and consequently antitumor efficacy[110,111].

CAR-engineered NK (CAR-NK) cell-based therapy has been developed pre-clinically to treat myeloid malignancies, targeting antigens such as CD38 and CD123[22,112,113]. CAR-NK cells have demonstrated potent anti-leukemia activity and high safety profile. Notably, these cells are highly efficient at countering tumor immune evasion by deploying various CAR-independent mechanisms, including NKR-mediated killing, along with antibody-dependent cell-mediated cytotoxicity (ADCC)[42,114]. The efficacy of CAR-NK cells in myeloid malignancy treatment is matched by their commendable safety profile, attributed to their high target specificity, which preserves hematopoiesis and minimizes epithelial tissue toxicity[22]. Moreover, the non-HLA-restricted modality of CAR-NK cells mitigates the GvHD risks, while a divergent cytokine profile partly reduces the incidence of CRS[74,115–117]. Despite progress, CAR-NK cell-based immunotherapy still faces challenges such as difficulties in cell expansion, cryopreservation, and maintaining long-term in vivo persistence[118–120]. Addressing these hurdles will be instrumental in advancing CAR-NK cell therapy as a cornerstone in myeloid malignancy management.

Given the limitations of current CAR-T and CAR-NK cell therapies, there is a critical need for a cell source that necessitates less genetic manipulation and offers enhanced safety. NKT cells emerge as an ideal candidate due to their innate-like properties and lack of GvHD risk.

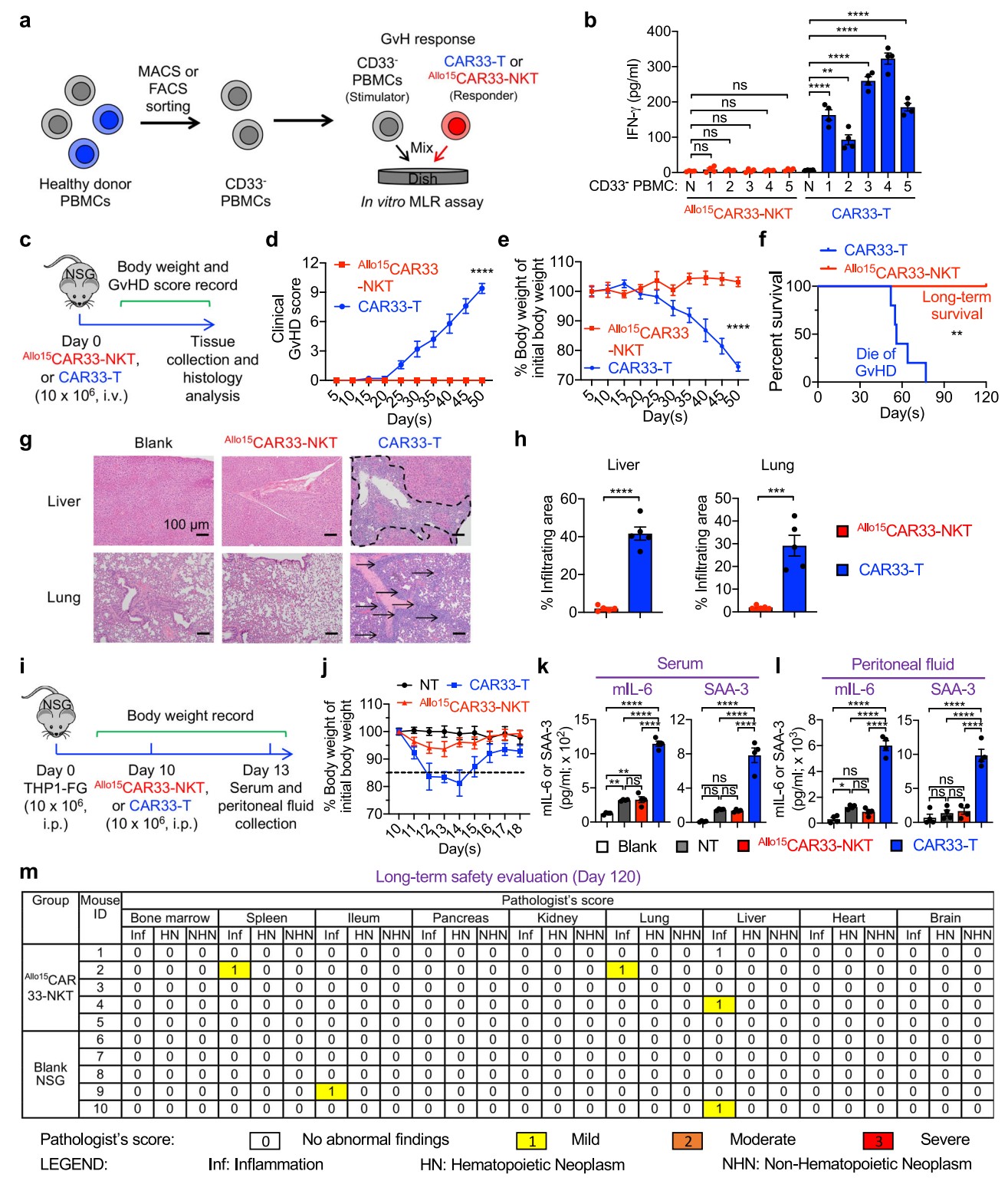

Previous clinical trials utilizing anti-GD2 CAR-NKT cells against neuroblastoma have demonstrated promising results regarding maximum tolerated dose and efficacy[58–60]. Additionally, a recent clinical trial reported on the use of off-the-shelf anti-CD19 CAR-NKT cells for treating relapsed or refractory B-cell malignancies[121]. This study demonstrated that these allogeneic CAR-NKT cells are well tolerated and can mediate objective responses in cancer patients, even at low doses[121]. However, allogeneic CAR-NKT therapy has not yet been applied to myeloid malignancies. To address this gap, we investigated the application of engineered NKT cells expressing CD33-directed CAR

to target AML and MDS, aiming for precise and comprehensive eradication of malignant cells through multiple mechanisms. Notably, our study employs the upregulation of CD1d expression on myeloblasts, providing an additional target for NKT TCR recognition (Fig. 1c–e). Additionally, NKT cells possess NKR-mediated killing capabilities, which align with the increased expression of NKR ligands, such as CD112, CD155, and MICA/B, observed on blasts and particularly LSPCs in both AML and MDS patients (Fig. 1c–e, j, k).

In this study, we report the development of BM-homing allogeneic CAR33-NKT cells for targeting BM-resident myeloid malignancies such

**Fig. 9 | Safety study of $^{Allo15}$CAR33-NKT cells. a,b** Studying the graft-versus-host (GvH) response of $^{Allo15}$CAR33-NKT cells using an in vitro mixed lymphocyte reaction (MLR) assay. CD33-negative PBMCs were pre-sorted using MACS or FACS and used as stimulator cells. Conventional CAR33-T cells were included as responder controls. **a** Experimental design. **b** ELISA analyses of IFN-γ production on day 4. N, no addition of stimulator cells ($n = 4$). **c–h** Studying the GvHD risk of $^{Allo15}$CAR33-NKT cells using a human xenograft NSG mouse model. **c** Experimental design. **d** Clinical GvHD score recorded over time ($n = 5$). The score was calculated as the sum of individual scores of 6 categories (body weight, activity, posture, skin thickening, diarrhea, and dishevelment; score 0–2 for each category). p was calculated using Day 50 data. **e** Body weight measured over time ($n = 5$). p was calculated using Day 50 data. **f** Kaplan–Meier survival curves ($n = 5$). **g** H&E-stained tissue sections. Tissues were collected from experimental mice on day 50. Scale bar, 100 μm. **h** Quantification of (**g**) ($n = 5$). **i–l** Studying the CRS response induced by $^{Allo15}$CAR33-NKT cells using a THP1-FG human AML xenograft NSG mouse model. **i** Experimental design. **j** Body weight of experimental mice over time ($n = 4$). ELISA analyses of mouse IL-6 and SAA3 in mouse serum (**k**) or peritoneal fluid (**l**) ($n = 4$). NT, samples collected from tumor-bearing mice receiving no therapeutic cell treatment. **m** Studying the long-term safety of $^{Allo15}$CAR33-NKT cells using a human xenograft NSG mouse model. Tissues from experimental mice were collected 120 days after injection with $^{Allo15}$CAR33-NKT cells. Data were presented as pathologist's scores of individual mouse tissues ($n = 5$). Representative of 1 (**k**) and 3 (**a–j**) experiments. Data are presented as the mean ± SEM. ns not significant, $*p < 0.05$; $**p < 0.01$; $***p < 0.001$; $****p < 0.0001$, by Student's $t$ test (**d**, **e**, **h**), one-way ANOVA (**b**, **k**, **l**), or log rank (Mantel-Cox) text adjusted for multiple comparisons (**f**). Source data and exact $p$ values are provided as a Source Data file.

as AML and MDS, leveraging human HSPC gene engineering and a clinically guided culture method. The resultant $^{Allo15}$CAR33-NKT cells exhibit a unique PK/PD profile showcasing effective and predominant homing to the BM, likely attributed to their intrinsic high expression of CXCR4 and CCR5 (Fig. 4), which are known to facilitate the homing of T and NK cells to the BM[80,81]. Different from approaches that boost BM migration by overexpressing CXCR4 on therapeutic cells like CAR-NK cells[122], $^{Allo15}$CAR33-NKT cells stably express high levels of BM-homing chemokine receptors, eliminating the need for further genetic modifications (Fig. 4h–j). The pronounced BM homing ability of these cells allows them to directly target malignant myeloid cells within their primary niche, a key factor in effective disease management and potential eradication of AML and MDS[13]. Additionally, their sustained activity within the BM bolsters their therapeutic efficacy, enabling prolonged cytotoxic effects and continuous immune surveillance, critical for treating these hematologic conditions (Figs. 4, 5, 6, and Supplementary Fig. 8).

Another significant challenge in treating myeloid malignancies is the presence of bone marrow-resident LSPCs, which are known for their roles in therapy resistance and immune evasion[4,12]. Targeting these LSPCs is crucial for the development of effective new therapies. In vivo studies demonstrate that $^{Allo15}$CAR33-NKT cells can migrate effectively to bone marrow tumor sites and eliminate tumor cells, showing a superior capacity compared to conventional CAR33-T cells (Figs. 5, 6, and Supplementary Fig. 8). Notably, in humanized mouse models, $^{Allo15}$CAR33-NKT cells can also target and eliminate CD33-negative/low LSPCs, a capability not observed with CAR33-T cells (Fig. 5a–n). Further, in vitro assays using primary samples from patients with AML or MDS have confirmed that $^{Allo15}$CAR33-NKT cells can effectively deplete LSPCs that express low levels of CD33, unlike CAR33-T cells, which fail to effectively target these cells (Fig. 5o–s). This unique ability of $^{Allo15}$CAR33-NKT cells may be attributed to their potent effector/memory characteristics, reduced exhaustion features, and their utilization of multiple tumor-targeting mechanisms (Fig. 3 and Supplementary Figs. 9, 10). These properties contribute to their enhanced efficacy in targeting the critical LSPC population within myeloid malignancies.

HMAs, including FDA-approved Azacitidine and Decitabine, are frontline treatments for myeloid malignancies[6,10]. HMAs function as cytidine nucleoside analogs, exerting their effects by inhibiting DNA methyltransferases and instigating DNA damage and cellular apoptosis during replication cycles. This process induces the modulation of gene expression, including the upregulation of specific genes, which can be identified as potential therapeutic targets[7,8]. Clinical trials have reported promising efficacy by combining HMAs with the CD70-specific monoclonal antibody Cusatuzumab, resulting in the elimination of LSCs and the induction of hematological responses in patients with AML, with 8 out of 12 patients achieving complete remission[24]. These results indicate the potential feasibility of combining HMAs with antigen-targeting therapies for the treatment of myeloid malignancies, encompassing monoclonal antibodies, antibody-drug conjugates (ADCs), and CAR-based cell therapy. In our study, although there was

no significant increase in CD33 expression observed, treatment of malignant cells with the HMA Decitabine yielded a significant upregulation of CD1d and NK ligands (Fig. 7a–c), indicating the potential of HMA combination therapy to enhance the effectiveness of NK and NKT cell-based therapies. Furthermore, increased antitumor capacity of $^{Allo15}$CAR33-NKT cells was observed both in vitro and in vivo with the combination of Decitabine (Fig. 7d–k, and Supplementary Fig. 12), suggesting the promise of the combination therapy as a potentially effective therapeutic approach for improving outcomes in patients with myeloid malignancies.

IL-15 has been employed in autologous CAR-NKT cell therapies in the treatment of neuroblastoma, showing safety and improved in vivo performance of CAR-NKT cells[58–60]. A recent phase I clinical trial investigating IL-15-enhanced GPC3-targeting CAR-T cells for the treatment of HCC demonstrated their antitumor capacity but also raised safety concerns due to an increased incidence of CRS[104]. This finding aligns with our preclinical studies, which indicate that $^{Allo15}$CAR33-NKT cells possess comparable in vivo antitumor efficacy but exhibit a significantly improved safety profile, as evidenced by reduced risks of CRS and xenogeneic GvHD (Fig. 10 and Supplementary Fig. 15). Consequently, careful consideration is warranted regarding the inclusion of IL-15 in conventional CAR-T cell therapies and the management of associated safety issues. Furthermore, a comprehensive investigation into the potential safety concerns related to IL-15-enhanced allogeneic CAR-NKT cells in clinical settings is crucial. Alongside IL-15, genes encoding immune-enhancing molecules such as IL-7, IL-12, IL-18, and IL-21, as well as immunosuppression-resistant elements like immune checkpoint inhibitors (e.g., anti-PD-1 antibody and dominant-negative TGF-β receptor), can be integrated to augment the antitumor efficacy and persistence of allogeneic CAR33-NKT cells[40,123].

Our data suggest that $^{Allo15}$CAR33-NKT cells minimally affect healthy hematopoietic cell populations, likely due to the absence or low expression of CD33 on HSCs (Fig. 8). However, the expression of CD33 on myeloid progenitor and mature myeloid cells raises concerns about potential myeloablation and resultant prolonged neutropenia, a life-threatening toxicity[124,125]. To mitigate this risk, the incorporation of suicide switch systems, such as sr39TK, inducible Cas9, and truncated EGFR, is able to reinforce the safety profile of $^{Allo15}$CAR33-NKT cells[126–128]. Furthermore, due to their allogeneic nature, $^{Allo15}$CAR33-NKT cells are likely to be rejected by host cells after the therapeutic window, similar to other allogeneic cell products[109,116,129,130], and the risk of long-term life-threatening toxicity from $^{Allo15}$CAR33-NKT cells could be limited.

## Methods
### Study approval
This study complies with all relevant ethical regulations. All experiments involving primary AML and MDS patient samples were approved by the Ronald Reagan UCLA Medical Center. Animal studies were approved by the Division of Laboratory Animal Medicine at UCLA. Healthy donor PBMCs were provided by the UCLA/CFAR Virology Core Laboratory without identification information under

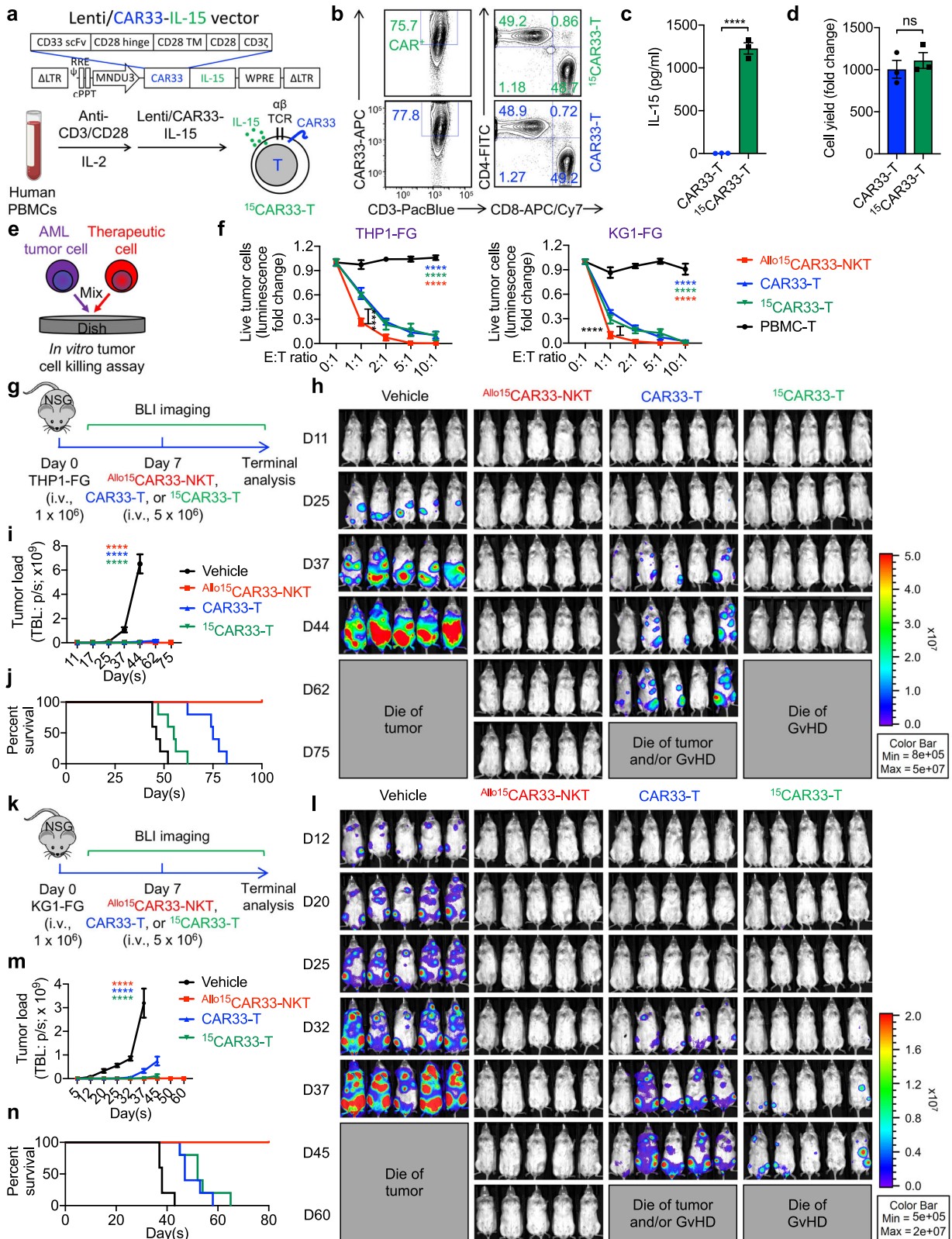

federal and state regulations. The PBMCs and peripheral blood units (HemaCare) were obtained from healthy donors who provided informed consent.

## Mice

NOD.Cg-Prkdc[SCID]Il2rg[tm1Wjl]/SzJ (NOD/SCID/IL-2Rγ[-/-], NSG) mice were maintained in the animal facilities of UCLA under the following

housing conditions: temperature ranging from 68 °F to 79 °F, humidity maintained at 30% to 70%, a light cycle of On at 6:00 am and Off at 6:00 pm, and room pressure set to negative. 6–10 weeks old male or female mice were used for all experiments unless otherwise indicated. Sex was not considered in the study design and analysis, as no significant differences were observed in the human AML NSG mouse models used. All animal experiments were approved by the

**Fig. 10 | Comparison of $^{Allo15}$CAR33-NKT cells and IL-15-enhanced conventional CAR33-T ($^{15}$CAR33-T) cells. a–d** Generation of $^{15}$CAR33-T cells. **a** Diagram showing the design of Lenti/CAR33-IL-15 lentivector, and the generation of $^{15}$CAR33-T cells from healthy donor PBMCs. Created in BioRender. LI, Y. (2025) https://BioRender.com/o37h997 **b** FACS detection of the CAR33 and CD4/CD8 co-receptors on CAR33-T and $^{15}$CAR33-T cells. **c** ELISA analyses of IL-15 production by CAR33-T and $^{15}$CAR33-T cells cultured in vitro for 24 h ($n = 3$; n indicates different PBMC donors). Note the successful incorporation and expression pf *IL-15* transgene in the $^{15}$CAR33-T cells. **d** Yield of CAR33-T and $^{15}$CAR33-T cells ($n = 3$; n indicates different PBMC donors). **e,f** Studying the antitumor efficacy of $^{15}$CAR33-T cells against human AML cell lines. **e** Experimental design. **f** Tumor cell killing data at 24 h ($n = 4$). **g–j** Studying the in vivo antitumor efficacy of $^{15}$CAR33-T cells using a THP1-FG

human AML xenograft NSG mouse model. **g** Experimental design. **h** BLI images showing the presence of tumor cells in experimental mice over time. **i** Quantification of (**h**) ($n = 5$). **j** Kaplan–Meier survival curves of experimental mice over time ($n = 5$). **k–n** Studying the in vivo antitumor efficacy of $^{15}$CAR33-T cells using a KG1-FG human AML xenograft NSG mouse model. **k** Experimental design. **l** BLI images showing the presence of tumor cells in experimental mice over time. **m** Quantification of (**l**) ($n = 5$). **n** Kaplan–Meier survival curves of experimental mice over time ($n = 5$). Note that the data for the Vehicle, $^{Allo15}$CAR33-NKT, and CAR33-T groups were also presented in the main Figs. 5h–k. Representative of 2 (**g–n**) and 3 (**a–f**) experiments. Data are presented as the mean ± SEM. ns, not significant; ****$p < 0.0001$, by Student's $t$ test (**c, d**), one-way ANOVA (**i, m**), or two-way ANOVA (**f**). Source data and exact $p$ values are provided as a Source Data file.

Institutional Animal Care and Use Committee of UCLA. All mice were bred and maintained under specific pathogen-free conditions, and all experiments were conducted in accordance with the animal care and use regulations of the Division of Laboratory Animal Medicine at the UCLA. Given the nature of the blood cancer models, there were no restrictions on tumor size or burden, making direct inferences from external measures unfeasible. Since the tumor cells expressed luciferase, we established a radiance threshold of $\geq 10^{10}$ photons/second per mouse as an upper surrogate limit. In addition, animals that experienced a 20% loss of their original body weight were euthanized. Experimental mice were randomly assigned to treatment groups to avoid statistically significant differences in the baseline tumor burden.

## Media and reagents
The X-VIVO 15 Serum-Free Hematopoietic Cell Medium (cat. no. 04-418Q) was purchased from Lonza. The StemSpan™ T Cell Generation Kit (cat. no. 09940), comprising the StemSpan™ SFEM II Medium (cat. no. 09605), the StemSpan™ Lymphoid Progenitor Expansion Supplement (cat. no. 09915), the StemSpan™ LPMS (cat. no. 09930), the StemSpan™ Lymphoid Progenitor Differentiation Coating Material (cat. no. 09925), and the ImmunoCult™ Human CD3/CD28/CD2 T Cell Activator (cat. no.10970), and MethoCult™H4330 Methycellulose-Based Medium (cat. no. 04330) were purchased from StemCell Technologies. The CTS™ OpTmizer™ T-Cell Expansion SFM (no phenol red, bottle format, cat. no. A3705001), the RPMI 1640 cell culture medium (cat. no. MT10040CV), and the DMEM cell culture medium (cat. no. MT10013CV) were purchased from Thermo Fisher Scientific. The CryoStor® Cell Cryopreservation Media CS10 (cat. no. C2874) and Iscove's Modified Dulbecco's Medium (cat. no. I3390) was purchased from MilliporeSigma. The homemade C10 medium was made of RPMI 1640 cell culture medium, supplemented with FBS (10% vol/vol), P/S/G (1% vol/vol), MEM NEAA (1% vol/vol), HEPES (10 mM), Sodium Pyruvate (1 mM), Beta-Mercaptoethanol (β-ME) (50 μM), and Normocin (100 μg/ml). The homemade D10 medium was made of DMEM supplemented with FBS (10% vol/vol), P/S/G (1% vol/vol), and Normocin (100 μg/ml). The homemade R10 medium was made of RPMI supplemented with FBS (10% vol/vol), P/S/G (1% vol/vol), and Normocin (100 μg/ml).

α-Galactosylceramide (αGC, KRN7000, cat. no. 867000) was purchased from Avanti Polar Lipids. Recombinant human IL-2 (cat. no. 200-02), IL-3 (cat. no. 200-03), IL-7 (cat. no. 200-07), IL-15 (cat. no. 200-15), IL-21 (cat. no. 200-21), IFN-γ (cat. no. 300-02), Flt3 ligand (Flt3L, cat. no. 300-19), macrophage colony stimulating factor (M-CSF, cat. no. 300-25), stem cell factor (SCF, cat. no. 300-07), and thrombopoietin (TPO, cat. no. 300-18) were purchased from Peprotech. Fetal Bovine Serum (FBS, lot no. 2087050) were purchased from Gibco and β-ME (cat. no. 1610710) were purchased from Bio-Rad. Penicillin-Streptomycin-Glutamine (P/S/G, cat. no. 10-378-016), MEM non-essential amino acids (NEAA, cat. no. 11-140-050), HEPES Buffer Solution (cat. no. 15630080), and Sodium Pyruvate (cat. no. 11360070) were purchased from Gibco. Normocin was purchased from Invivogen (cat. no. NC9390718). Decitabine (Tocris Bioscience™, cat. no. 26-241-0) was purchased from Fisher Scientific.

## Lentiviral vectors
A parental lentivector, pMNDW, was utilized to construct the lentiviral vectors employed in this study[55]. The 2 A sequences derived from foot-and-mouth disease virus (F2A), porcine teschovirus-1 (P2A), and thosea asigna virus (T2A) were used to link the inserted genes to achieve co-expression.

The Lenti/iNKT-IL-15 vector was constructed by inserting into the pMNDW parental vector a synthetic tricistronic gene encoding human iNKT TCRα-F2A-iNKT TCRβ-P2A-IL-15 (IL-15 indicates the secreted form of human IL-15). The Lenti/iNKT-CAR33-IL-15 vector was constructed by inserting into the pMNDW parental vector a synthetic tetracistronic gene encoding human iNKT TCRα-F2A-iNKT TCRβ-P2A-CAR33-T2A-IL-15 (CAR33 indicates a CD33-directed CAR)[57]. The Lenti/CAR33 vector was constructed by inserting into the pMNDW parental vector a synthetic gene encoding CAR33. The Lenti/CAR33-IL-15 vector was constructed by inserting into the pMNDW parental vector a synthetic bicistronic gene encoding CAR33-T2A-IL-15. The Lenti/FG vector was constructed by inserting into pMNDW parental vector a synthetic bicistronic gene encoding Fluc-P2A-EGFP. The Lenti/CD33 vector was constructed by inserting into pMNDW parental vector a synthetic gene encoding human CD33. Synthetic gene fragments were sourced from GenScript (Piscataway, NJ, USA) and IDT (Coralville, IA, USA). Lentiviral particles were generated utilizing HEK 293 T cells by employing a standardized transfection procedure with the Trans-IT-Lenti Transfection Reagent (Mirus Bio). Subsequently, a concentration protocol was applied using Amicon™ Ultra Centrifugal Filter Units in accordance with the manufacturer's specifications (MilliporeSigma).

## Cell lines
Human AML cell line THP1 (cat. no. TIB-202), KG1 (cat. no. CCL-246), and HL60 (cat. no. CCL-240) were purchased from the American Type Culture Collection (ATCC). To establish stable tumor cell lines that overexpress both firefly luciferase and enhanced green fluorescent protein dual reporters (FG), the parental tumor cell lines were transduced with lentiviral vectors carrying the specific genes of interest (i.e., Lenti/FG). 72 h after lentiviral transduction, the cells underwent flow cytometry sorting to isolate the genetically modified cells (as identified as GFP⁺ cells) necessary for creating stable cell lines. This process resulted in the development of three stable tumor cell lines for this study: THP1-FG, KG1-FG, and HL60-FG. The THP1-FG$^{CD33-/-}$ cell line was generated by knocking out the *CD33* gene from the parental THP1-FG cell line using CRISPR/Cas9. The THP1-FG$^{CD1d-/-}$ cell line was generated by knocking out the *CD1d* gene from the parental THP1-FG cell line. The THP1-FG$^{CD33/CD1d-/-}$ cell line was generated by knocking out the *CD1d* gene from the THP1-FG$^{CD33-/-}$ cell line using CRISPR/Cas9. The single guide RNAs (sgRNAs) targeting the *CD33* gene (GGGGAGUUCUU-GUCGUAGUA) and *CD1d* gene (GCUUUACCUCCCGGUUUAAG) was purchased from Synthego, and was introduced into AML tumor cells via electroporation using an Amaxa 4D Nucleofection X Unit (Lonza), according to the manufacturer's instructions.

The artificial antigen presenting cell line (aAPC) was generated by engineering the K562 human chronic myelogenous leukemia cell line

(ATCC, cat. no. CCL-243) to overexpress human CD80/CD83/CD86/4-1BBL co-stimulatory receptors[44]. The aAPC-CD33 cell lines were generated by further engineering the parental aAPC line to overexpress human CD33.

All tumor cell lines utilized in this study underwent short tandem repeat (STR) profiling, and the resulting profiles were compared to established databases to confirm accurate identification. Furthermore, the cell lines were regularly screened for mycoplasma contamination to preserve their integrity and authenticity.

### Human CD34+ hematopoietic stem and progenitor cells (HSPCs), periphery blood mononuclear cells (PBMCs), and primary patient bone marrow samples

Purified human CD34+ HSPCs derived from cord blood (CB) were acquired from HemaCare. granulocyte-colony stimulating factor (G-CSF)-mobilized peripheral blood units from healthy donors were sourced from either HemaCare or Cincinnati Children's Hospital Medical Center (CCHMC), after which CD34+ HSPCs were isolated using magnetic-activated cell sorting with a CliniMACS device (Miltenyi Biotec, cat. no. 220-002-126), following the manufacturer's guidelines. For all isolations, the CD34+ cell purity exceeded 97%, as confirmed by flow cytometry. Healthy donor PBMCs were provided by the UCLA/CFAR Virology Core Laboratory without identification information under federal and state regulations. Primary AML and MDS patient bone marrow samples were collected at the Ronald Reagan UCLA Medical Center from consented patients through an IRB-approved protocol (IRB#22-000558) and processed. Information regarding the patients' gender and age was not provided in this study to avoid including three or more indirect identifiers for the study participants. Patient gender was not considered in the study design and was determined based on self-reporting.

### Antibodies and flow cytometry

Fluorochrome-conjugated antibodies specific for human CD45 (Clone HI30, PerCP, FITC, or Pacific Blue-conjugated, 1:500, cat. no. 982318, 982316, or 982306), TCR αβ (Clone IP26, Pacific Blue or PE-Cy7-conjugated, 1:25, cat. no. 306716 or 306720), CD3 (Clone HIT3a, Pacific Blue, PE, or PE-Cy7-conjugated, 1:500, cat. no. 300330, 300308, or 300316), CD4 (Clone OKT4, PE-Cy7, PerCP or FITC-conjugated, 1:500, cat. no. 317414, 317432 or 317408), CD8 (Clone SK1, PE, APC-Cy7, or APC-conjugated, 1:300, cat. no. 344706, 344714 or 344722), CD8β (Clone QA20A40, PE-Cy7-conjugated, 1:500, cat. no. 376708), CD14 (Clone HCD14, Pacific Blue-conjugated, 1:100, cat. no. 367122), CD15 (Clone W6D3, APC-conjugated, 1:500, cat. no. 323008), CD19 (Clone HIB19, APC-Cy7-conjugated, 1:200, cat. no. 302218), CD20 (Clone 2H7, APC-Cy7-conjugated, 1:200, cat. no. 302314), CD33 (Clone WM53, APC or PE-Cy7-conjugated, 1:100, cat. no. 983902 or 983908), CD34 (Clone 581, PerCP- conjugated, 1:500, cat. no. 343520), CD38 (Clone HIT2, Pacific Blue-conjugated, 1:200, cat. no. 980316), CD56 (Clone HCD56, FITC or PerCP-conjugated, 1:10, cat. no. 318304 or 318342), CD69 (Clone FN50, PE-Cy7 or PerCP-conjugated, 1:50, cat. no. 310912 or 310928), CD1d (Clone 51.1, PE-Cy7 or APC-conjugated, 1:50, cat. no. 350310 or 350308), CD112 (Clone TX31, PE-conjugated, 1:250, cat. no. 337410), CD155 (Clone SKII.4, PE-Cy7-conjugated, 1:250, cat. no. 337614), CD11b (Clone ICRF44, FITC-conjugated, 1:500, cat. no. 982614), MICA/MICB (Clone 6D4, PE or APC-conjugated, 1:25, cat. no. 320906 or 320908), 41BBL (Clone 5F4, PE-conjugated, 1:500, cat. no. 311504), CD83 (Clone HB15e, APC-Cy7-conjugated, 1:500, cat. no. 305330), CD86 (Clone IT2.2, APC-conjugated, 1:500, cat. no. 305412), PD-1 (Clone A17188A, PE or FITC-conjugated, 1:25, cat. no. 379210 or 379206), TIM-3 (Clone A18087E, APC-conjugated, 1:25, cat. no. 364804), LAG-3 (Clone 7H2C65, PE-Cy7-conjugated, 1:25, cat. no. 369208), NKG2D (Clone 1D11, PE-Cy7-conjugated, 1:50, cat. no. 320812), DNAM-1 (Clone 11A8, APC-conjugated, 1:50, cat. no. 338312), NKp30 (Clone P30-15, APC-conjugated, 1:50, cat. no. 325210), NKp46

(Clone 9E2, FITC-conjugated, 1:50, cat. no. 137606), CXCR4 (Clone 12G5, PE-Cy7-conjugated, 1:50, cat.no. 306514), CCR5 (Clone J418F1, APC-Cy7-conjugated, 1:50, cat. no. 359110), SOX2 (Clone 14A6A34, Pacific Blue-conjugated, 1:200, cat. no. 656112), OCT3/4 (Clone 3A2A20, PE-conjugated, 1:200, cat. no. 653704), IFN-γ (Clone B27, PE-Cy7-conjugated, 1:50, cat. no. 506518), Granzyme B (Clone QA16A02, APC-conjugated, 1:2000 or 1:5000, cat. no. 372204), Perforin (Clone dG9, PE-Cy7-conjugated, 1:50 or 1:100, cat. no. 308126), TNF-α (Clone MAb11, APC-conjugated, 1:4000, cat. no. 502912), IL-2 (Clone MQ1-17H12, APC-Cy7-conjugated, 1:50, cat. no. 500342), β2-microglobulin (B2M) (Clone 2M2, FITC or APC-conjugated, 1:2000 or 1:5000, cat. no. 316304 or 316311), HLA-DR (Clone L243, APC-Cy7-conjugated, 1:200 or 1:500, cat. no. 307618), and HLA-DR, DP, DQ (Clone Tü39, FITC-conjugated, 1:200 or 1:500, cat. no. 361706) were purchased from BioLegend. Fluorochrome-conjugated antibodies specific for human iNKT TCR Vα24-Jβ18 (Clone 6B11, PE-conjugated, 1:20, cat. no. 552825) were purchased from BD Biosciences. Fluorochrome-conjugated antibody specific for human iNKT TCR Vβ11 (Clone C21, APC-conjugated, 1:50, cat. no. IM2290) was purchased from Beckman-Coulter. Fluorochrome-conjugated antibodies specific for human ULBP-1 (Clone 170818, PE-conjugated or unconjugated, 1:25, cat. no. FAB1380P or MAB1380) and ULBP-2,5,6 (Clone 165903, APC-conjugated, 1:25, cat. no. FAB1298A) were purchased from R&D Systems. A goat anti-mouse IgG F(ab')2 secondary antibody was purchased from ThermoFisher (cat. no. A-11001). Fixable Viability Dye eFluor506 (e506, 1:500, cat. no. 65-0866-14) was purchased from Affymetrix eBioscience; mouse Fc Block (anti-mouse CD16/32, cat. no. 553141) was purchased from BD Biosciences; and human Fc Receptor Blocking Solution (TrueStain FcX) was purchased from BioLegend (cat. no. 422302). In our study, note the use of antibodies with identical clones but differing conjugated fluorochromes, with one typical antibody listed herein.

All flow cytometry staining was performed following standard protocols, as well as specific instructions provided by the manufacturer of a particular antibody. Appropriate isotype staining controls were used for all staining procedures. Stained cells were analyzed using a MACSQuant Analyzer 10 flow cytometer (Miltenyi Biotech), following the manufacturer's instructions. FlowJo software version 9 (BD Biosciences) was used for data analysis.

### Enzyme-linked immunosorbent cytokine assays (ELISAs)

The ELISAs for measuring human and mouse cytokines were conducted according to a standard protocol provided by BD Biosciences[55]. Supernatants from cell culture experiments were collected and analyzed to quantify cytokines (e.g., human IFN-γ). The capture and biotinylated antibodies used for cytokine detection were sourced from BD Biosciences, while the streptavidin-HRP conjugate was obtained from Invitrogen. Human cytokine standards were purchased from eBioscience, and the Tetramethylbenzidine (TMB) substrate was acquired from Thermo Scientific (cat. no. PI34021). The quantification of mouse IL-6 was performed using a paired purified anti-mouse IL-6 antibody (Clone MP5-20F3, cat. no. 504501) and a biotinylated anti-mouse IL-6 antibody (Clone MP5-32C11, cat. no. 504601) from BioLegend. Mouse SAA-3 levels were determined using a Mouse SAA-3 ELISA Kit (MilliporeSigma, cat. no. EZMSAA3-12K), following the manufacturer's instructions. Absorbance of the samples was measured at 450 nm using an Infinite M1000 microplate reader (Tecan).

### Generation of HSPC-engineered allogeneic IL-15-enhanced CD33-directed CAR-engineered NKT (Allo15CAR33-NKT) cells

Allo15CAR33-NKT cells were generated by differentiating gene-engineered human cord blood CD34+ HSPCs in a 5-stage feeder-free Ex Vivo HSPC-Derived NKT Cell Culture method[44]. At Stage 0, the frozen stock of human CD34+ HSPCs was thawed and resuscitated in T cell X-VIVO 15 Serum-Free Hematopoietic Stem Cell Medium

supplemented with human Flt3L (50 ng/ml), SCF (50 ng/ml), TPO (50 ng/ml), and IL-3 (20 ng/ml) for 24 h. The cells then underwent lentiviral transduction for another 24 h via two systems adapted from an established protocol: a) a one-vector system or (b) a two-vector system[55,56].

a) The one-vector system. A single transgene plasmid, composed of iNKT TCR, CAR33, and the secreting form of human IL-15 cytokine, was inserted into the parental pMNDW vector, resulting in one vector: Lenti/iNKT-CAR33-IL15. The human CD34$^+$ HSPCs were then transfected with the single vector Lenti/iNKT-CAR33-IL15.

b) The two-vector system. Two transgene plasmids were constructed, one composed of iNKT TCR and the secreting form of human IL-15 cytokine, while the other composed of CAR33. These transgene plasmids were respectively inserted into the parental pMNDW vector, resulting in two vectors: Lenti/iNKT-IL15 and Lenti/CAR33. The human CD34$^+$ HSPCs were then transfected with both vectors simultaneously.

At Stage 1, gene-engineered HSPCs harvested from Stage 0 were cultured in the feeder-free StemSpan™ SFEM II Medium supplemented with StemSpan™ Lymphoid Progenitor Expansion Supplement for ~14 days. HSPCs were cultured in CELLSTAR®24-well Cell Culture Nontreated Multiwell Plates (VWR, cat. no. 82050-892). StemSpan™ Lymphoid Differentiation Coating Material (500 μl/well, diluted to a final concentration of 1X from a stock dilution of 100X) was applied to the plates and left for 2 h at room temperature or overnight at 4 °C. Subsequently, 500 μl of the transduced CD34$^+$ HSPC suspension, with a density of $2 \times 10^4$ cells/ml, was added to each pre-coated well. Half of the medium in each well was removed and replaced with fresh medium twice per week.

At Stage 2, the Stage 1 cells were harvested and cultured in the feeder-free StemSpan™ SFEM II Medium supplemented with StemSpan™ LPMS for ~7 days. StemSpan™ Lymphoid Differentiation Coating Material (1 ml/well, diluted to a final concentration of 1X) was applied to Non-Treated Falcon™ Polystyrene 6-well Microplates (Thermo Fisher Scientific, cat. no. 140675); 2 ml of the harvested Stage 1 cells, resuspended with a density of $1 \times 10^5$ cells/ml, was added into each pre-coated well. The cell density was maintained at $1-2 \times 10^6$ cells per well during the Stage 2 culturing. Cells were passaged 2–3 times per week with the addition of fresh medium for each passage.

At Stage 3, the Stage 2 cells were harvested and cultured in the feeder-free StemSpan™ SFEM II Medium supplemented with StemSpan™ LPMS, CD3/CD28/CD2 T Cell Activator, and human recombinant IL-15 (20 ng/ml) for ~7 days. StemSpan™ Lymphoid Differentiation Coating Material (1 ml/well, diluted to a final concentration of 1X) was applied to Non-Treated Falcon™ Polystyrene 6-well Microplates (Thermo Fisher Scientific, cat. no. 08-772-49); 2 ml of the harvested Stage 2 cells, resuspended with a density of $5 \times 10^5$ cells/ml, was added into each pre-coated well. The cell density was maintained at $1-2 \times 10^6$ cells per well during the Stage 3 culturing. Cells were passaged 2–3 times per week with the addition of fresh medium for each passage.

At Stage 4, the Stage 3 cells were harvested and verified by flow-cytometry to confirm their status as mature $^{Allo15}$CAR33-NKT cells or their derivatives; then the cells underwent expansion stage via the following three approaches: (a) an αCD3/αCD28 antibody-based expansion, (b) an αGC/PBMC-based expansion, or (c) an artificial APC (aAPC)-based expansion. This expansion process spanned ~7–14 days and could be carried out in the homemade C10 medium, or the feeder-free, serum-free CTS™ OpTmizer™ T-Cell Expansion SFM (Thermo Fisher Scientific). Following the manufacturer's instructions, the expanded $^{Allo15}$CAR33-NKT cells were aliquoted and cryopreserved in CryoStor® Cell Cryopreservation Media CS10 using a Thermo Scientific™ CryoMed™ Controlled-Rate Freezer 7450 (Thermo scientific) for stock.

a) The αCD3/αCD28 antibody-based expansion. Ultra-LEAF™ Purified Anti-Human CD3 Antibody (Clone OKT3; BioLegend, cat. no.

317347) was coated onto CELLSTAR®24-well Cell Culture Nontreated Multiwell Plates (VWR) at a concentration of 1 μg/ml (500 μl/well) and allowed to sit for 2 h at room temperature or overnight at 4 °C. Expansion medium supplemented with IL-7 (10 ng/ml), IL-15 (10 ng/ml), and Ultra-LEAF™ Purified Anti-Human CD28 antibody (1 μg/ml) (Clone CD28.2; BioLegend, cat. no. 302943) was used to resuspend mature $^{Allo15}$CAR33-NKT cells harvested from the Stage 3 culturing at a density of $5 \times 10^5$ cells/ml; 2 ml of the cell suspension was then added to each pre-coated well. After 3 days of culturing, cells were harvested and resuspended in a fresh expansion medium supplemented with IL-7 (10 ng/ml) and IL-15 (10 ng/ml) at a density of $0.5-1 \times 10^6$ cells/ml; 2 ml of the cell suspension was added into each well of Corning™ Costar™ Flat Bottom Cell Culture 6-well Plates (Corning, cat. no. 3516; no αCD3 antibody coating). The cell density was maintained at $0.5-1 \times 10^6$ cells/ml during the expansion stage. Cells were passaged 2-3 times per week with the addition of fresh for each passage.

b) The αGC/PBMC approach. An established protocol was followed to load healthy donor PBMCs with α-Galactosylceramide (5 μg/ml; Avanti Polar Lipids) in C10 medium for 1 h, followed by 6000 rads irradiation using Rad Source RS-2000 X-Ray Irradiator (Rad Source Technologies). The Stage 3 mature $^{Allo15}$CAR33-NKT cells and derivatives were co-cultured with the irradiated αGC/PBMCs with a ratio of 1:5. The mixed cells were resuspended in expansion medium supplemented with human IL-7 (10 ng/ml) and IL-15 (10 ng/ml) at a density of $0.5-1 \times 10^6$ cells/ml; 2 ml of the cell suspension was seeded into each well of Corning™ Costar™ Flat Bottom Cell Culture 6-well Plates. The cell density was maintained at $0.5-1 \times 10^6$ cells/ml during the expansion stage. Cells were passaged 2–3 times per week with the addition of fresh medium for each passage.

c) The aAPC expansion approach. aAPCs were irradiated at 10,000 rads using a Rad Source RS-2000 X-Ray Irradiator (Rad Source Technologies). The Stage 3 mature $^{Allo15}$CAR33-NKT cells and derivatives were co-cultured with the irradiated aAPCs (with a ratio of 1:1. The cells were resuspended in expansion medium supplemented with human IL-7 (10 ng/ml) and IL-15 (10 ng/ml) at a density of $0.5-1 \times 10^6$ cells/ml; 2 ml cell suspension was seeded into each well of the Corning™ Costar™ Flat Bottom Cell Culture 6-well Plates. The cell density was maintained at $0.5-1 \times 10^6$ cells/ml during the expansion stage. Cells were passaged 2-3 times per week with the addition of fresh medium for each passage.

### Generation of PBMC-derived conventional αβ T (PBMC-T)

PBMCs from healthy donors were utilized to generate conventional αβ T cells, referred to as PBMC-T cells. To produce PBMC-T cells, PBMCs were activated using Dynabeads™ Human T-Activator CD3/CD28 (Thermo Fisher Scientific, cat. no. 11131D) following the manufacturer's guidelines. The activated cells were then cultured in C10 medium supplemented with 20 ng/ml IL-2 for a duration of 2 to 3 weeks.

### Generation of CD33-directed CAR-engineered conventional αβ T (CAR33-T) cells and their IL-15-enhanced derivatives

PBMCs from healthy donors were utilized to generate CAR33-engineered conventional αβ T cells and their IL-15-enhanced derivatives, referred to as CAR33-T and $^{15}$CAR33-T cells, respectively. To produce these cells, non-treated tissue culture 24-well plates (Corning, cat. no. 3738) were coated with Ultra-LEAF™ Purified Anti-Human CD3 Antibody (Clone OKT3, BioLegend) at 1 μg/ml (500 μl/well), at room temperature for 2 h or at 4 °C overnight. PBMCs were resuspended in the C10 medium supplemented with 1 μg/ml Ultra-LEAF™ Purified Anti-Human CD28 Antibody (Clone CD28.2, BioLegend) and 30 ng/ml IL-2, followed by seeding in the pre-coated plates at $1 \times 10^6$ cells/ml (1 ml/well). After 2 days, the cells were transduced with either Lenti/CAR33 or Lenti/CAR33-IL-15 viruses for a period of 24 h. The resultant CAR33-T and $^{15}$CAR33-T cells were expanded for about 2 weeks in C10 medium and then cryopreserved for future applications.

### In vitro tumor cell killing assay

Various human AML tumor cells (i.e., THP1-FG, KG1-FG, HL60-FG, THP1-FG$^{CD33-/-}$, THP1-FG$^{CD1d-/-}$, and THP1-FG$^{CD33/CD1d-/-}$ cells; $1 \times 10^4$ cells per well) were co-cultured with the indicated therapeutic cells (i.e., PBMC-T, CAR33-T, $^{15}$CAR33-T, and $^{Allo15}$CAR33-NKT cells) in Corning 96-well clear bottom black plates for 24 h, in C10 medium with or without the addition of αGC (100 ng/ml). The E:T ratio is indicated in the figure legends. At the end of culture, viable AML tumor cells were quantified by adding D-luciferin (150 µg/ml; Fisher Scientific, cat. no. 50-209-8110) to cell cultures, followed by the measurement of luciferase activity using an Infinite M1000 microplate reader (Tecan). In some experiments, 10 µg/ml Ultra-LEAF™ purified anti-human NKG2D (Clone 1D11, BioLegend, cat. no. 320813) or anti-human DNAM-1 antibody (Clone 11A8, BioLegend, cat. no. 338302) was added to co-cultures to investigate the mechanism of tumor cell killing mediated by NKRs, and LEAF™ purified mouse IgG2bk isotype control antibody (Clone MG2b-57, BioLegend, cat. no. 401202) was included as an isotype control.

### In vitro Decitabine treatment assay

Human AML cell line THP1 was cultured in C10 medium for 2 weeks, with or without the addition of Decitabine. Decitabine was supplemented into the cell culture every two days at a concentration of 2 µM. At the end of the cell culture, THP1 cells were harvested and subjected to further analysis, including flow cytometry and in vitro tumor cell killing assays.

### In vitro assays using AML and MDS patient samples

Primary AML and MDS patient BM samples were collected and subsequently diluted in PBS and subjected to density gradient centrifugation using Ficoll-Paque (Thermofisher Scientific) to obtain mononuclear cells following the manufacturer's instructions. The resulting cells were cryopreserved for future use. In one assay, the primary AML and MDS patient samples were analyzed for tumor/blast cell phenotype using flow cytometry. Blast cells were identified as SSC$^{low}$CD45$^{med}$ cells, and 3 subpopulations were further identified using CD34 and CD38 markers. Specifically, LSPCs were identified as SSC$^{low}$CD45$^{med}$CD34$^+$CD38$^-$ cells, CMPs were identified as SSC$^{low}$CD45$^{med}$CD34$^+$CD38$^+$ cells, and MBCs were identified as SSC$^{low}$CD45$^{med}$CD34$^-$CD38$^+$ cells. Surface expression of CD33, CD1d, and NK ligands (i.e., CD112, CD155, and MICA/B) on the subpopulations of blast cells were analyzed using flow cytometry. In another assay, the primary AML and MDS patient samples were used to study tumor/blast cell killing by $^{Allo15}$CAR33-NKT cells. Blast cells were sorted by FACS sorting, followed by co-culturing with various therapeutic cells (E:T ratio 1:1) in C10 medium in Corning 96-well Round Bottom Cell Culture plates for 24 h. At the end of culture, cells were collected and live blast cells (identified as CD3$^-$6B11$^-$ cells) was analyzed using flow cytometry.

### In vitro assays using healthy donor PBMC and CD34$^+$ HSPC samples

Healthy donor PBMCs and CD34$^+$ HSPCs were analyzed using flow cytometry. Monocytes were identified as CD11b$^+$CD14$^+$CD3$^-$ cells, granulocytes were identified as CD11b$^+$CD15$^+$CD3$^-$ cells, B cells were identified as CD19$^+$CD3$^-$ cells, NK cells were identified as CD56$^+$CD3$^-$ cells, T cells were identified as CD3$^+$ cells, CD4 T cells were identified as CD3$^+$CD4$^+$ cells, CD8 T cells were identified as CD3$^+$CD8$^+$ cells. The expression of surface markers (i.e., CD33 and CD1d) on each cell types were analyzed using flow cytometry. In the in vitro immune cell killing assay, PBMCs and CD34$^+$ HSPCs were co-cultured with the therapeutic cells (E:T ratio 1:1) in C10 medium in Corning 96-well Round Bottom Cell Culture plates for 24 h. At the end of culture, cells were collected, and live target cells was analyzed using flow cytometry.

### In vitro HSPC colony formation assay

Healthy donor CD34$^+$ HSPCs were thawed and co-cultured with $^{Allo15}$CAR33-NKT or CAR33-T cells at a 1:1 ratio in StemSpan™ Serum-Free Expansion Medium (StemCell Technologies, cat. no. 09650) at a concentration of 10,000 HSPCs per ml for 4 h. Following co-culture, the CD34$^+$ HSPCs were purified by MACS sorting using a Human Pan T Cell Isolation Kit (Miltenyi Biotec, cat. no. 130-096-535), and then resuspended in Iscove's Modified Dulbecco's Medium with 2% FBS at a concentration of 5000 cells/ml. 150 CD34$^+$ HSPCs were dispensed into 300 µl MethoCult™H4330 Methylcellulose-Based Medium (StemCell Technologies, cat. no. 04330) supplemented with recombinant human IL-3 (20 ng/ml), SCF (50 ng/ml) and M-CSF (50 ng/ml) in a 24-well plate. After 10–14 days, Burst-Forming Unit-Erythroid (BFU-E) and Colony-Forming Unit-Granulocyte/Macrophage (CFU-GM) colonies were counted based on colony morphology and imaged using an Olympus IX70 Inverted Fluorescence Microscope (Olympus Life Science).

### In vitro mixed lymphocyte reaction (MLR) assay: studying GvH response

PBMCs from multiple healthy donors (>5) were pre-sorted to remove CD33-positive cells using MACS or FACS, irradiated at 2,500 rads, and subsequently utilized as stimulators to assess the GvH response of $^{Allo15}$CAR33-NKT cells as responders. PBMC-derived conventional CAR33-T cells served as a responder control. The stimulators ($5 \times 10^5$ cells/well) and responders ($2 \times 10^4$ cells/well) were co-cultured in 96-well round-bottom plates in C10 medium for a duration of 4 days. Following the culture period, the supernatants were collected and analyzed for IFN-γ production via ELISA.

### In vitro MLR assay: studying T cell-mediated allorejection

PBMCs from multiple healthy donors (>5) were utilized as responders to assess host T cell-mediated allorejection of $^{Allo15}$CAR33-NKT cells, which acted as stimulators. Conventional CAR33-T cells were included as a stimulator control p. Both $^{Allo15}$CAR33-NKT and CAR33-T cells were irradiated at a dose of 2500 rads. The stimulator cells ($5 \times 10^5$ per well) and responder cells ($2 \times 10^4$ per well) were co-cultured in 96-well round-bottom plates in C10 medium for a duration of 4 days. Following the culture period, the supernatants were collected and analyzed for IFN-γ production via ELISA.

### In vitro MLR assay: studying NK cell-mediated allorejection

To investigate host NK cell-mediated allorejection of $^{Allo15}$CAR33-NKT cells, PBMC-derived NK (PBMC-NK) cells from multiple healthy donors (>5) were utilized. Conventional CAR33-T cells were included as an allogeneic target cell control. PBMC-NK cells ($2 \times 10^4$ cells/well) were cocultured with their respective allogeneic target cells ($2 \times 10^4$ cells/well) in 96-well round-bottom plates using C10 medium for 24 h. Following the culture period, the cultures were harvested, and the number of live cells was quantified using flow cytometry.

### Immunostaining and fluorescent microscopy

During the development of $^{Allo15}$CAR33-NKT cells, cells were collected weekly. These cells were washed with PBS and blocked with 10% normal goat serum (ThermoFisher, cat. no. 50062Z). Subsequently, these were stained using the following antibodies: FITC anti-human CD3 antibody (OKT3, Biolegend, 1:50, cat. no. 317306) and PE anti-human iNKT TCR antibody (6B11, BD Biosciences, 1:2, cat. no. 342904) in 10% normal goat serum. After staining, the cells were washed with PBS, fixed using 4% PFA in PBS, washed with PBS, placed onto glass slides, and sealed using ProLong™ Gold Antifade reagent with DAPI (ThermoFisher, cat. no. P36931). For imaging, an Olympus IX70 fluorescence microscope was employed. The acquired images were subsequently processed in ImageJ to combine the color channels.

### In vivo bioluminescence live animal imaging (BLI)

BLI was conducted using the Spectral Advanced Molecular Imaging (AMI) HTX system (Spectral Instrument Imaging). Live animal images were captured 5 min after intraperitoneal (i.p.) administration of D-

Luciferin, with doses of 1 mg/mouse for tumor cell visualization and 3 mg/mouse for therapeutic cell visualization. The imaging data were processed and analyzed using AURA imaging software (Spectral Instrument Imaging, version 3.2.0).

### In vivo PK/PD study of Allo15CAR33-NKT/FG cells
Experimental design is shown in Fig. 4a. Briefly, on Day 0, NSG mice received intravenous (i.v.) inoculation of $^{Allo15}$CAR33-NKT/FG cells ($10 \times 10^6$ CAR$^+$ cells in 100 µl PBS per mouse), or control CAR33-T/FG cells ($10 \times 10^6$ CAR$^+$ cells in 100 µl PBS per mouse). Over the experiment, mice were monitored for survival and the therapeutic cells were measured using BLI. Note in this study, therapeutic immune cells, but not the tumor cells, were labeled with FG.

### In vivo antitumor efficacy study of $^{Allo15}$CAR33-NKT cells: human AML xenograft NSG mouse model
Experimental design is shown in Figs. 5a, h,and Supplementary Fig. 8a, e. Briefly, on Day 0, NSG mice received intravenous (i.v.) inoculation of human AML cells, including THP1-FG, KG1-FG, HL60-FG, and THP1-FG$^{CD33-/-}$ cells ($1 \times 10^6$ cells per mouse). On Day 7, the experimental mice received i.v. injection of vehicle (100 µl PBS per mouse), $^{Allo15}$CAR33-NKT cells ($10 \times 10^6$ CAR$^+$ cells in 100 µl PBS per mouse), or control CAR33-T cells ($10 \times 10^6$ CAR$^+$ cells in 100 µl PBS per mouse). Over the experiment, mice were monitored for survival and their tumor loads were measured twice per week using BLI.

### In vivo antitumor efficacy study of $^{Allo15}$CAR33-NKT cells: human AML patient-derived xenograft (PDX) NSG mouse model
Experimental design is shown in Fig. 6a. Briefly, NSG mice were subjected to 300 rads of total body irradiation and subsequently intravenously infused with $2 \times 10^6$ bone marrow blast cells from AML patients (in 100 µl of PBS with 0.5% FBS). After a six-week period, following assessment of engraftment levels, mice received intravenous injection of vehicle (100 µl PBS per mouse), $^{Allo15}$CAR33-NKT cells ($10 \times 10^6$ CAR$^+$ cells in 100 µl PBS per mouse), or control CAR33-T cells ($10 \times 10^6$ CAR$^+$ cells in 100 µl PBS per mouse). Over the experiment, mice were monitored for survival. At the end of the experiment, mice were euthanized, and their tumor burdens and presence of human therapeutic cells were assessed using flow cytometry and immunohistochemistry (IHC) staining.

### In vivo combination therapy study of $^{Allo15}$CAR33-NKT cells with Decitabine using a human AML xenograft NSG mouse model
Experimental design is shown in Fig. 7h and Supplementary Fig. 12a. Briefly, on Day 0, NSG mice received i.v. inoculation of human THP1-FG or KG1-FG cells ($1 \times 10^6$ cells per mouse). On Day 3, the experimental mice received intraperitoneal (i.p.) injection of Decitabine (0.2 mg in 100 µl PBS per mouse). On Day 10, the experimental mice received i.v. injection of vehicle (100 µl PBS per mouse) or $^{Allo15}$CAR33-NKT cells ($10 \times 10^6$ CAR$^+$ cells in 100 µl PBS per mouse). Over the experiment, mice were monitored for survival and their tumor loads were measured twice per week using BLI.

### In vivo safety study of $^{Allo15}$CAR33-NKT cells on hematopoietic precursors: BLT humanized mouse model
BLT (human bone marrow-liver-thymus engrafted NSG mice) humanized mice were generated as previously described, with some modifications[55,91]. In brief, human CD34$^+$ HSPCs were thawed and intravenously injected into NSG mice (-0.5–1 × 10$^6$ cells per recipient) that had received 270 rads of total body irradiation. 1–2 fragments of human fetal or postnatal thymus (-1 mm$^3$) were implanted under the kidney capsule of each recipient NSG mouse. The mice were maintained on trimethoprim/sulfmethoxazole (TMS) chow in a sterile environment for 8 weeks until analysis or use for further experiments. 8 weeks following HSPC engraftment, mice underwent quality checking for human leukocyte reconstitution using flow cytometry. Only mice with a minimum of ≥25% human CD45$^+$ cells were selected for use in the experiments. These BLT humanized mice received intravenous infusion of $10 \times 10^6$ $^{Allo15}$CAR33-NKT cells, and CAR33-T cells were included as a control. On Day 7 post-infusion, the mice were euthanized, and samples from bone marrow, spleen, and peripheral blood were analyzed using flow cytometry to assess changes in multilineage engraftment.

### In vivo graft-versus-host disease (GvHD) evaluation
Experimental design is shown in Fig. 9c. Briefly, on Day 0, NSG mice received intravenously (i.v.) injection of $^{Allo15}$CAR33-NKT cells ($10 \times 10^6$ CAR$^+$ cells in 100 µl PBS per mouse), or control CAR33-T cells ($10 \times 10^6$ CAR$^+$ cells in 100 µl PBS per mouse). Over the experiment, mice were monitored for survival and their body weight and GvHD score were measured. A score ranging from 0 to 2 was assigned for each clinical GvHD sign, which includes body weight, activity, posture, skin thickening, diarrhea, and dishevelment[83]. At the end of the experiment, multiple tissues were collected and prepared for histological analysis. Note that this model primarily addresses xenogeneic GvHD, which we refer to simply as GvHD throughout this study.

### In vivo cytokine release syndrome (CRS) evaluation
Experimental design is shown in Fig. 9i and follows the design from a prior study[99,102]. Briefly, on Day 0, NSG mice received intraperitoneally (i.p.) inoculation of THP1-FG cells ($10 \times 10^6$ cells per mouse). On Day 10, the experimental mice received i.p. injection of vehicle (100 µl PBS per mouse), $^{Allo15}$CAR33-NKT cells ($10 \times 10^6$ CAR$^+$ cells in 100 µl PBS per mouse), or control CAR33-T cells ($10 \times 10^6$ CAR$^+$ cells in 100 µl PBS per mouse). On Days 13, blood and peritoneal fluid samples were collected from the experimental mice, and their serum IL-6 and SAA-3 were measured using ELISA. A Mouse SAA-3 ELISA Kit (Millipore Sigma) was used to measure SAA-3, following the manufacturer's instructions.

### In vivo organ damage evaluation
On Day 0, NSG mice received intravenously (i.v.) injection of vehicle (100 µl PBS per mouse), or $^{Allo15}$CAR33-NKT cells ($10 \times 10^6$ CAR$^+$ cells in 100 µl PBS per mouse). On Day 120, various mouse tissues were collected and analyzed by the UCLA Pathology Core. Tissues were analyzed for inflammation (Inf), hematopoietic neoplasm (HN), and non-hematopoietic neoplasm (NHN). Data were presented as pathologist's scores of individual mouse tissues ($n = 9$–$10$). 0, no abnormal findings; 1, mild; 2, moderate; 3, severe.

### Single cell RNA sequencing (scRNA-seq)
In one study, scRNA-seq was utilized to examine the gene profiles of primary AML and MDS patient-derived malignant blast cells, as shown in Figs. 1a, f–k, and Supplementary Fig. 1b–d. Data from Gene Expression Omnibus database (GSE235923) and NCBI Sequence Read Archive (PRJNA720840) were included for scRNA-seq analyses[49,50].

In another study, scRNA-seq was utilized to examine the gene profiles of therapeutic cells with or without tumor cell coculture, as shown in Supplementary Fig. 10a. In the culture of both therapeutic cell types ($^{Allo15}$CAR33-NKT and CAR33-T cells), THP1-FG tumor cells were introduced daily at an E:T ratio of 1:1 to simulate tumor cell rechallenge. After 10 days of coculture, therapeutic cells were harvested and sorted via FACS for the e506 (viability dye)$^-$CD3$^+$ population, followed by scRNA-seq. Note that in our study, the same $^{Allo15}$CAR33-NKT and CAR33-T cell samples (prior to the THP1-FG tumor cell challenge) were prepared for 5' GEX + TCR Library sequencing, and analyzed using both scRNA-seq and scTCR-seq.

The following methods for scRNA-seq was performed as described previously[44], specifically within the "Methods, scRNA-seq" section.

Freshly collected samples were immediately delivered to the UCLA TCGB Core for library construction and scRNA-seq. Cells were

quantified using a Cell Countess II automated cell counter (Invitrogen/Thermo Fisher Scientific). A total of 10,000 cells from each experimental group were loaded on the Chromium platform (10X Genomics), and libraries were constructed using the Chromium Next GEM Single Cell 3' Kit or 5' Kit and the Chromium Next GEM Chip G Single Cell Kit (10X Genomics), according to the manufacturer's instructions. Library quality was assessed using the D1000 ScreenTape on a 4200 TapeStation System (Agilent Technologies). Libraries were sequenced on an Illumina NovaSeq using the NovaSeq S4 Reagent Kit (100 cycles; Illumina).

For cell clustering and annotation, the merged digital expression matrix generated by Cellranger was analyzed using an R package Seurat (v.4.0.0) following the guidelines[131–133]. Briefly, after filtering the low-quality cells, the expression matrix was normalized using NormalizeData function, followed by selecting variable features across datasets using FindVariableFeatures and SelectIntegrationFeatures functions. To correct the batch effect, FindIntegrationAnchors and IntegrateData functions were used based on the selected feature genes. The corrected dataset was subjected to standard Seurat workflow for dimension reduction and clustering. In this study, clusters of therapeutic cells were manually merged and annotated based on gene signatures reported from Human Protein Atlas (proteinatlas.org) and previous studies[134–138] and clusters of mouse immune cells were merged and annotated based on the immune lineage markers. AddModuleScore was used to calculate module scores of each list of gene signatures, and FeaturePlot function was used to visualize the expression of each signature in the UMAP plots. For gene set enrichment analysis (GSEA), clusterProfiler packages[139,140] were used to calculate the enrichment scores of each cluster in the signature gene list.

Pseudotime trajectories were analyzed using the R package Monocle3 (v.2.18.0)[141]. After clustering analysis, the data dimensionality of the intended cell types was reduced using UMAP. Monocle was utilized to learn trajectory through cluster_cells function by the default method. Next, a principal graph was fit within each partition using the learn_graph function and trajectories with numerous branches were reconstructed. After specifying the root nodes of the intended trajectory, the order_cells function was used to calculate where each cell falled in pseudotime of the biological process. Finally, the plot_cells function was utilized to show the trajectory graph in UMAP and to color cells by pseudotime. Heatmaps showing the bifurcation expression patterns were generated using function plot_genes_branched_heatmap with DEGs with the FDR adjusted $P < 1 \times 10^{-50}$.

The gene expression matrix for certain cell types was extracted with single cell IDs and the gene names. The databases were uploaded to the CytoTRACE website (https://cytotrace.stanford.edu/) to predict differentiation states. CytoTRACE score of each cell was calculated and integrated back into the scRNA-seq database and plotted on the UMAP afterward. 2D graphics were generated utilizing the FeaturePlot function.

### Single cell TCR sequencing (scTCR-seq)

$^{Allo15}$CAR33-NKT (6B11$^+$TCRαβ$^+$) and CAR33-T (6B11$^-$TCRαβ$^+$) cells were sorted using a FACSAria II flow cytometer. Sorted cells were immediately delivered to the UCLA TCGB Core to perform scTCR-seq using a 10X Genomics Chromium™ Controller Single Cell Sequencing System (10X Genomics), following the manufacturer's instructions and the TCGB Core's standard protocol. Briefly, the cells were encapsulated in gel beads and lysed to release RNA, which was then reverse-transcribed and amplified. The resulting cDNA was fragmented and barcoded, and 5' GEX + TCR libraries were constructed using the Illumina Tru-Seq RNA Sample Prep Kit (Cat#FC-122-1001). The raw sequencing data were processed and mapped to the human TCR reference genome (hg38) using the Cell Ranger VDJ software. The frequencies of the α or β chain recombination were plotted to identify the unique TCR repertoires in each cell population. Note that in our study, the same

$^{Allo15}$CAR33-NKT and CAR33-T cell samples (prior to the THP1-FG tumor cell challenge) were prepared for 5' GEX + TCR Library sequencing, and analyzed using both scRNA-seq and scTCR-seq.

### Histology

Mouse issues were collected, fixed in 10% Neutral Buffered Formalin for up to 36 h, and embedded in paraffin for sectioning (5 μm thickness). Tissue sections were prepared and stained with Hematoxylin and Eosin (H&E) by the UCLA Translational Pathology Core Laboratory (TPCL), following the Core's standard protocols. Stained sections were imaged using an Olympus BX51 upright microscope equipped with an Optronics Macrofire CCD camera (AU Optronics). The images were analyzed using an Optronics PictureFrame software (AU Optronics).

### Statistics

Graphpad Prism 8 software (Graphpad) was used for statistical data analysis. Student's two-tailed $t$ test was used for pairwise comparisons. Ordinary 1-way ANOVA followed by Tukey's or Dunnett's multiple comparisons test was used for multiple comparisons. Log rank (Mantel-Cox) test adjusted for multiple comparisons was used for Meier survival curves analysis. Data are presented as the mean ± SEM, unless otherwise indicated. In all figures and figure legends, "n" represents the number of samples or animals used in the indicated experiments. A P value of less than 0.05 was considered significant. ns, not significant.

### Reporting summary

Further information on research design is available in the Nature Portfolio Reporting Summary linked to this article.

## Data availability

The scRNA-seq and scTCR-seq data generated in this study (related to Fig. 2h and Supplementary Figs. 10 and 11) have been deposited in the public repository Gene Expression Omnibus Database: https://www.ncbi.nlm.nih.gov/geo/query/acc.cgi?acc=GSE270430. The scRNA-seq publicly available data used in this study (related to Fig. 1f–k and Supplementary Fig. 1b–d) are available in the Gene Expression Omnibus database (GSE235923; https://www.ncbi.nlm.nih.gov/geo/query/acc.cgi?acc=GSE235923) and NCBI Sequence Read Archive (PRJNA720840; https://www.ncbi.nlm.nih.gov/sra/?term=PRJNA720840). The publicly available data of CD33 mRNA expression used in this study (related to Supplementary Fig. 13a) are available in the Human Protein Atlas (https://www.proteinatlas.org/ENSG00000105383-CD33). The remaining data are available within the Article, Supplementary Information or Source Data file. Source data are provided with this paper.

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

## Acknowledgements

We thank the University of California, Los Angeles (UCLA) animal facility for providing animal support; the UCLA Translational Pathology Core Laboratory (TPCL) for providing histology support; the UCLA Technology Centre for Genomics & Bioinformatics (TCGB) facility for providing RNA-seq services; the UCLA CFAR Virology Core for providing human cells; and the UCLA BSCRC Flow Cytometry Core Facility for cell sorting support. This work was supported by a Partnering Opportunity for Discovery Stage Research Projects Award and a Partnering Opportunity for Translational Research Projects Award from the California Institute for Regenerative Medicine (DISC2-11157 and TRAN1-12250, to L.Y.), a Department of Defense CDMRP PRCRP Impact Award (CA200456 to L.Y.), a UCLA BSCRC Innovation Award (to L.Y.), and an Ablon Scholars Award (to L.Y.). Y.-R.L. is a postdoctoral fellow supported by a UCLA MIMG M. John Pickett Post-Doctoral Fellow Award, a CIRM-BSCRC Postdoctoral Fellowship, a UCLA Sydney Finegold Postdoctoral Award, and a UCLA Chancellor's Award for Postdoctoral Research. E.Z. is a postdoctoral fellow supported by a T32 UCLA/Caltech Integrated Cardiometabolic Medicine for Bioengineers fellowship (NIH/NHLBI T32HL144449). Some figures were created with BioRender (biorender.com).

## Author contributions

Y.-R.L., Y.F., S.N., Y.Z., and L.Y. designed the experiments, analyzed the data, and wrote the manuscript. L.Y. conceived and oversaw the study, with assistance from Y.-R.L., Y.F., S.N., and Y.Z., and suggestions from T.H. and P.W. Y.-R.L., Y.F., S.N., and Y.Z. performed all experiments, with assistance from Y.C., Z.L., E.Z., Y.T., and J.H. V.R. and S.K. helped with the generation of BLT mice. J.J.Z. helped with the statistical analysis of data. W.C.-H., S.P., C.S.S., and C.O. provided AML and MDS patient samples through an IRB-approved protocol.

## Competing interests

Y.-R.L., P.W., and L.Y. are inventors on the patent (Title: Engineered off-the-shelf immune cells and methods of use thereof; identification number: AU2020291457A1) relating to this study filed by UCLA. C.S.S. is a cofounder and stockholder of Pluto Immunotherapeutics. P.W. is a cofounder, stockholder and advisory board member of Simnova Bio, TCRCure Biopharma, Appia Bio, and is a scientific advisor to Grit Biotechnology. L.Y. is a scientific advisor to AlzChem and Amberstone Biosciences, and a co-founder, stockholder, and advisory board member of Appia Bio. Appia Bio licensed some patents relating to this study from UCLA. None of the declared companies contributed to or directed any of the research reported in this article. The remaining authors declare no competing interests.
