## [Transparent Peer Review file · Nature Communications]

Allogeneic CD33-directed CAR-NKT cells for the treatment of bone marrow-resident myeloid malignancies

Corresponding Author: Professor Lili Yang

This manuscript has been previously reviewed at another journal. This document only contains information relating to versions considered at Nature Communications. Mentions of previous journals have been redacted.

Version 0:

Reviewer comments:

Reviewer #1

(Remarks to the Author)

The authors have adequately addressed my concerns.

Reviewer #2

(Remarks to the Author)

Li and colleagues have submitted a revised manuscript to the one previously submitted to [journal name redacted] and now under consideration at Nature Communications. For the avoidance of doubts, I will be exclusively commenting on the aspects raised by previous reviewer #2. The authors have made a very credible efforts to address all points I have previously raised and importantly by generating or at least providing novel and additional data of high importance to this story line. While major aspects of my previous concerns (adequate controls, additional models), they have not been properly implemented in the MS, likely because of length restrictions. I have the following remaining concerns that require editing of the MS but not new data:

- 1) The addition of the CD33 CAR T control engineered with IL-15 and demonstration of superior features is essential to the manuscript. It should be accompanied by a careful discussion of the in vivo effects driven by xenoGVHD but with not dissimilar efficacy. In other words the potential for an autologous approach should be discussed too.
- 2) Additional in vivo efficacy data of the engineered NKT is equally important, particularly the PDX models. This would permit broadening of efficacy claims and take the concern of reproducibility and model dependency.
- 3) My challenge to the relevance of XENO-GVHD of human cells in mice to ALLO-GVHD in humans remains. I do not challenge that NK or NKT cells are not expected to trigger GVHD, as shown but the NSG are clearly overrepresenting the problem due to xenoreactivity. While I agree that the MLR data is supportive, this is the only ALLO-human data. This must be carefully discussed in the MS and claims on these must be tuned down.
- 4) Because the argument of CD33 expression by HSC (or lack) thereof will be controversially seen in the field. The investigators must carefully balance and discuss these findings against other expectations from literature.
- 5) The reading of the legends is still difficult in some instances. For example figure 1e mentions n =4 same for 3e, etc, but it is unclear what this means.
- 6) Overall if space is required to meet journal requirements the current discussion have certainly substantial room for reduction (and thereby permit space for the points above), as a lot of space is used up to paraphrase or repeat results. Similarly, some figures may be moved to the supplements or removed entirely. For example I am unsure about the value of figure 6, which is descriptive only.

Reviewer #3

(Remarks to the Author)

In this revised manuscript, the authors have responded appropriately to the reviewer comments, highlighting novel aspects of their manuscript as compared with their previous studies.

Li and colleagues have submitted a revised manuscript to the one previously submitted to [journal name redacted] and now under consideration at Nature Communications. For the avoidance of doubts, I will be exclusively commenting on the aspects raised by previous reviewer #2. The authors have made a very credible efforts to address all points I have previously raised and importantly by generating or at least providing novel and additional data of high importance to this story line. While major aspects of my previous concerns (adequate controls, additional models), they have not been properly implemented in the MS, likely because of length restrictions. I have the following remaining concerns that require editing of the MS but not new data:

1) The addition of the CD33 CAR T control engineered with IL-15 and demonstration of superior features is essential to the manuscript. It should be accompanied by a careful discussion of the *in vivo* effects driven by xenoGVHD but with not dissimilar efficacy. In other words the potential for an autologous approach should be discussed too.

Response: We thank the reviewer for their valuable suggestions. We have moved these data to the main figure, included the result description, and incorporated the following discussion in the revised manuscript to address these points (See revised manuscript, **Figure 10** and **Page 29**).

“IL-15 has been employed in autologous CAR-NKT cell therapies in the treatment of neuroblastoma, showing safety and improved *in vivo* performance of CAR-NKT cells^{60,122,123}. A recent phase I clinical trial investigating IL-15-enhanced GPC3-targeting CAR-T cells for the treatment of HCC demonstrated their antitumor capacity but also raised safety concerns due to an increased incidence of CRS¹⁰⁴. This finding aligns with our preclinical studies, which indicate that Allo¹⁵CAR33-NKT cells possess comparable *in vivo* antitumor efficacy but exhibit a significantly improved safety profile, as evidenced by reduced risks of CRS and xenogeneic GvHD (**Fig. 10** and **Supplementary Fig. 15**). Consequently, careful consideration is warranted regarding the inclusion of IL-15 in conventional CAR-T cell therapies and the management of associated safety issues. Furthermore, a comprehensive investigation into the potential safety concerns related to IL-15-enhanced allogeneic CAR-NKT cells in clinical settings is crucial.”

2) Additional *in vivo* efficacy data of the engineered NKT is equally important, particularly the PDX models. This would permit broadening of efficacy claims and take the concern of reproducibility and model dependency.

Response: We appreciate the reviewer’s suggestions and have moved these data to the main **Figure 6**.

3) My challenge to the relevance of XENO-GVHD of human cells in mice to ALLO-GVHD in humans remains. I do not challenge that NK or NKT cells are not expected to triggered GVHD, as shown but the NSG are clearly overrepresenting the problem due to xenoreactivity. While I agree that the MLR data is supportive, this is the only ALLO-human data. This must be carefully discussed in the MS and claims on these must be tuned down.

Response: We appreciate the reviewer's suggestions and acknowledge that the GvHD observed in the NSG model primarily pertains to xenogeneic GvHD. We have revised the manuscript to clarify this point, particularly changing the terminology to "xenogeneic GvHD."

4) Because the argument of CD33 expression by HSC (or lack) thereof will be controversially seen in the field. The investigators must carefully balance and discuss these findings against other expectations from literature.

Response: We thank the reviewer and included the discussion in the revised manuscript:

"Our data suggest that ^{Allo}CAR33-NKT cells minimally affect healthy hematopoietic cell populations, likely due to the absence or low expression of CD33 on HSCs (Fig. 8). However, the expression of CD33 on myeloid progenitor and mature myeloid cells raises concerns about potential myeloablation and resultant prolonged neutropenia, a life-threatening toxicity^{129,130}. To mitigate this risk, the incorporation of suicide switch systems, such as sr39TK, inducible Cas9, and truncated EGFR, is able to reinforce the safety profile of ^{Allo}CAR33-NKT cells^{131–133}. Furthermore, due to their allogeneic nature, ^{Allo}CAR33-NKT cells are likely to be rejected by host cells after the therapeutic window, similar to other allogeneic cell products^{110,117,134,135}, and the risk of long-term life-threatening toxicity from ^{Allo}CAR33-NKT cells could be limited."

5) The reading of the legends is still difficult in some instances. For example figure 1e mentions n =4 same for 3e, etc, but it is unclear what this means.

Response: We have revised the figure legends and clarified the means of n.

6) Overall if space is required to meet journal requirements the current discussion have certainly substantial room for reduction (and thereby permit space for the points above), as a lot of space is used up to paraphrase or repeat results. Similarly, some figures may be moved to the supplements or removed entirely. For example I am unsure about the value of figure 6, which is descriptive only.

Response: We appreciate the reviewer's suggestions and have modified the figure display accordingly. Specifically, we have included three main figures and relocated the previous Figure 6 to the supplementary figures section.